# Reversible promoter methylation determines fluctuating expression of acute phase proteins

**Shi-Chao Zhang[1†], Ming-Yu Wang[1†], Jun-Rui Feng[2†], Yue Chang[1], Shang-Rong Ji[1]\*, Yi Wu[2]\***

[1]MOE Key Laboratory of Cell Activities and Stress Adaptations, School of Life Sciences, Lanzhou University, Lanzhou, China; [2]MOE Key Laboratory of Environment and Genes Related to Diseases, School of Basic Medical Sciences, Xi'an Jiaotong University, Xi'an, China

**Abstract** Acute phase reactants (APRs) are secretory proteins exhibiting large expression changes in response to proinflammatory cytokines. Here we show that the expression pattern of a major human APR, that is *C-reactive protein* (*CRP*), is casually determined by DNMT3A and TET2-tuned promoter methylation status. *CRP* features a CpG-poor promoter with its CpG motifs located in binding sites of STAT3, C/EBP-β and NF-κB. These motifs are highly methylated at the resting state, but undergo STAT3- and NF-κB-dependent demethylation upon cytokine stimulation, leading to markedly enhanced recruitment of C/EBP-β that boosts *CRP* expression. Withdrawal of cytokines, by contrast, results in a rapid recovery of promoter methylation and termination of *CRP* induction. Further analysis suggests that reversible methylation also regulates the expression of highly inducible genes carrying CpG-poor promoters with APRs as representatives. Therefore, these CpG-poor promoters may evolve CpG-containing TF binding sites to harness dynamic methylation for prompt and reversible responses.

**\*For correspondence:**
jsr@lzu.edu.cn (S-RJ);
wuy@lzu.edu.cn (YW)

[†]These authors contributed equally to this work

**Competing interests:** The authors declare that no competing interests exist.

## Introduction

Acute phase reactants (APRs) are liver-produced plasma proteins constituting an integral part of innate defense (*Gabay and Kushner, 1999*; *Medzhitov, 2007*). They are defined by a substantial change (>25%) of their plasma concentrations in response to inflammation. IL-6 (*Kopf et al., 1994*) and IL-1β (*Zheng et al., 1995*) are chief inducers of APR expression through activation of STAT3, NF-κB and C/EBP in hepatocytes (*Bode et al., 2012*; *Quinton et al., 2012*; *Poli, 1998*). C-reactive protein (CRP) is the first APR to be discovered, whose plasma concentrations at baseline are less than 2–3 μg/ml, but can rapidly increase up to 1000-fold upon infection or tissue injury; the heightened levels of CRP, however, return to the baseline with the resolution of inflammation (*Pepys and Hirschfield, 2003*; *Du Clos, 2013*; *Pathak and Agrawal, 2019*). The mechanisms of *CRP* induction have been thoroughly examined by reporter assays and truncation analysis. A region of ~220 bp in the proximal promoter of *CRP* that contains (nonconical) binding sites for STAT3, NF-κB and C/EBP-β is identified to be sufficient to mediate *CRP* induction by IL-6 and IL-1β (*Singh et al., 2007*; *Young et al., 2008*; *Figure 1A*).

Intriguingly, a promoter SNP (rs3091244) associated with plasma levels of CRP is located at 286 bp upstream the transcription start site (*Szalai et al., 2005*; *Zacho et al., 2008*; *Allin et al., 2010*). This SNP does not exist in binding sites of transcription factors (TFs) critical to *CRP* expression. Rather, the major −286C allele constitutes a CpG motif, at which DNA methylation frequently occurs; whereas the minor alleles of −286A/T disrupt the CpG motif and are associated with enhanced *CRP* expression. Beside the −286CpG, there are only four additional CpGs within the

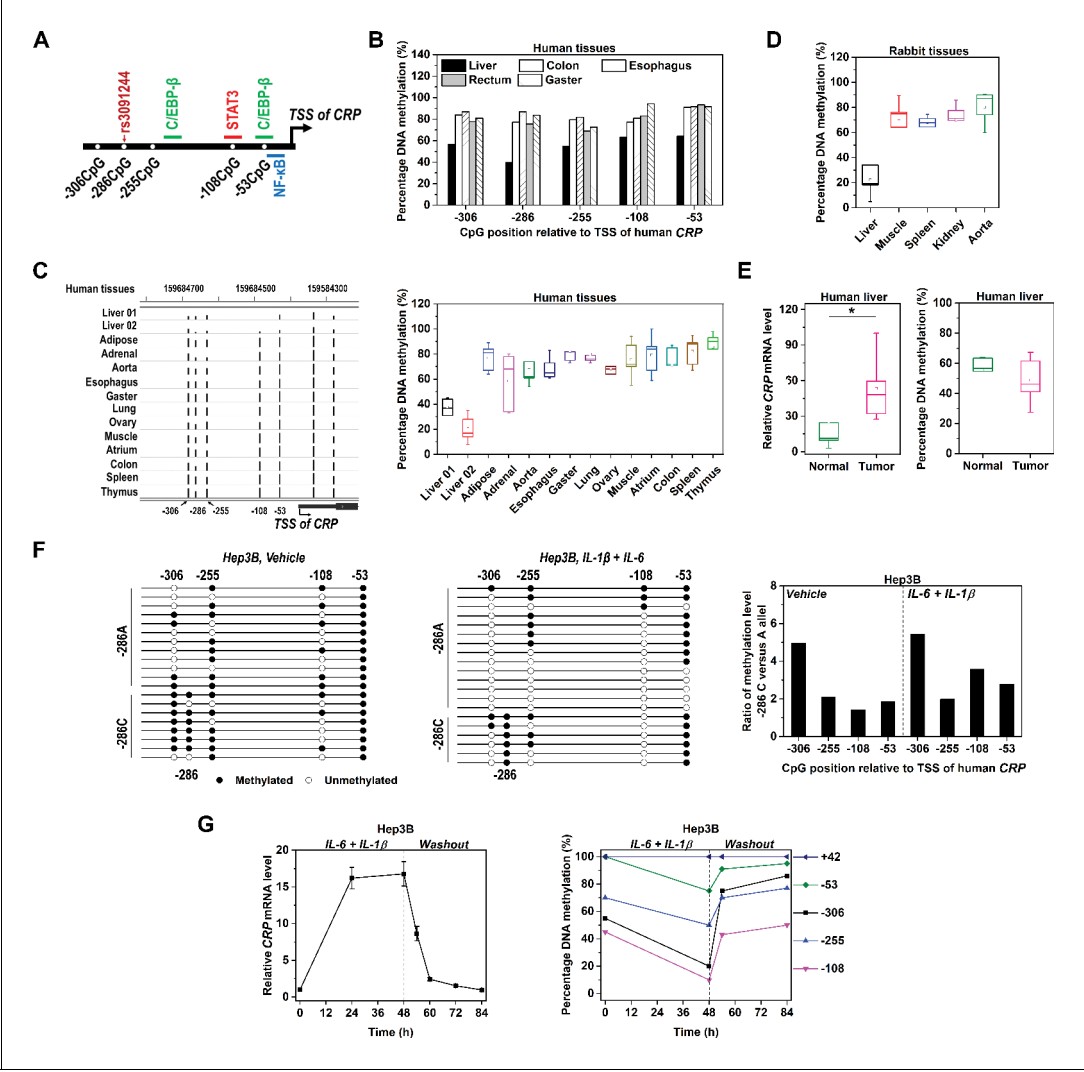

**Figure 1.** Methylation level of *CRP* promoter is inversely associated with expression. (**A**) Schematic illustration of *CRP* promoter, in which SNP rs3091244, CpG motifs and TF binding sites are indicated. (**B**) Methylation levels of *CRP* promoter (−550 ~ 1 bp) in pooled normal human tissues adjacent to tumors (five liver, eight colon, 10 esophagus, 10 rectum and 10 gaster) were determined by bisulfite cloning sequencing. (**C**) Methylation levels of *CRP* promoter in normal human tissues were retrieved from available GEO datasets generated by whole-genome bisulfite sequencing: Liver 01-GSM916049, Liver 02-GSM1716965, Adipose-GSM1120331, Adrenal-GSM1120325, Aorta-GSM1120329, Esophagus-GSM983649, Gaster-GSM1120333, Lung-GSM983647, Ovary-GSM1120323, Muscle-GSM1010986, Atrium-GSM1120335, Colon-GSM983645, Spleen-GSM983652, Thymus-GSM1120322 (Gene Expression Omnibus database). The bisulfite sequencing tracks of *CRP* promoter (left; the height of the black bars represents percentage of DNA methylation) and pooled analysis (right) are shown. (**D**) Methylation levels of *CRP* promoter in rabbit tissues were determined by bisulfite cloning sequencing. Liver is the major organ expressing *CRP* in both humans and rabbits. Accordingly, the methylation levels of *CRP* promoter are lower in normal liver tissues than in other tissues. (**E**) Levels of *CRP* expression (left) and promoter methylation (right) in tumor versus normal tissues from human livers (n = 5) were determined by q-PCR and bisulfite cloning sequencing, respectively. Liver tumors exhibit higher levels of *CRP* expression but lower levels of promoter methylation than adjacent normal liver tissues. *p<0.05 (paired t-test). (**F**) Bisulfite cloning sequencing of −286C versus −286A alleles of *CRP* promoter in Hep3B cells at resting (Vehicle treated) or induced states (IL-6 and IL-1β treated). −286A allele was less methylated than −286C allele at both states. (**G**) IL-6 (10 ng/ml) and IL-1β (1 ng/ml) treatment induced *CRP* expression (left) and promoter demethylation (right) in Hep3B cells, while withdraw of these cytokines led to a quick drop of *CRP* expression and promoter re-methylation. The result of one representative experiment is shown.

proximal promoter of *CRP*. Importantly, two of those CpGs are located at the binding sites of STAT3 and NF-κB/C/EBP-β (*Figure 1A*). Given that promoter methylation affects TF recruitment (*Hu et al., 2013*; *Yin et al., 2017*) and contributes to gene silencing (*Jones, 2012*; *Wu and Zhang, 2014*; *Dor and Cedar, 2018*; *Luo et al., 2018*; *Blattler and Farnham, 2013*), it is notable that levels of promoter methylation and expression of *CRP* appear to be negatively associated albeit with

undefined causality (*Wang et al., 2014*). In the present study, we demonstrate that the expression pattern of *CRP* is causally determined by reversible promoter methylation, and that this regulation may also apply to highly inducible genes with CpG-poor promoters.

## Results

### Promoter methylation is inversely associated with CRP expression

To determine whether promoter methylation affects *CRP* expression, we first compared methylation levels of *CRP* promoter in different human tissues. *CRP* is expressed predominantly, if not solely, by the liver (*Pepys and Hirschfield, 2003*; *Du Clos, 2013*). Accordingly, methylation levels of *CRP* promoter in normal liver tissues were much lower than that in other tissues (*Figure 1B*). Analysis of published bisulfite sequencing datasets also confirmed that *CRP* promoter was most demethylated in the liver (*Figure 1C*). Similar results were further obtained in rabbits (*Figure 1D*), wherein *CRP* exhibits a comparable expression pattern as in humans. Moreover, malignant liver tissues expressed more *CRP* than adjacent normal tissues, and they were also less methylated at *CRP* promoter (*Figure 1E*). These data together reveal an inverse association between levels of promoter methylation and *CRP* expression across different tissues or cell types.

Hepatic Hep3B cell line is a conventional model to investigate APR expression (*Singh et al., 2007*; *Young et al., 2008*). *CRP* promoter in Hep3B cells harbors distinct alleles at the −286 position, with −286C on one allele and −286A on the other. Intriguingly, in addition to lacking the −286CpG, all other promoter CpGs on the −286A allele were much less methylated than that on the −286C allele at the resting state (*Figure 1F*). Such an allelic imbalance of promoter methylation was further reinforced at the induced state. Notably, the induction of *CRP* by IL-6 and IL-1β was accompanied by prominent promoter demethylation (*Figure 1G*). Following washout of the cytokines, however, both the expression and the promoter methylation of *CRP* were rapidly recovered. By contrast, the methylation level of a 5′ UTR CpG remained constant during the entire time course. Therefore, levels of promoter methylation and *CRP* expression are also specifically and dynamically associated in the same cell type.

### Promoter methylation causally determines CRP expression

To clarify whether the observed association is causal, we directly modulated methylation levels of *CRP* promoter and examined its consequence on expression. Treating Hep3B cells with 5-aza or RG108 to inhibit DNA methylation significantly enhanced *CRP* expression at the resting state, but showed little effect at the induced state (*Figure 2A*; *Figure 2—figure supplement 1*) wherein *CRP* promoter also underwent active demethylation (*Figure 1G*). Nevertheless, 5-aza could moderately rescue the induced expression of CRP when STAT3 or NF-κB was inhibited (*Figure 2B and C*), hinting for their involvement in active demethylation of *CRP* promoter. Moreover, in vitro methylation before transfection markedly suppressed the reporter activity of *CRP* promoter in Hep3B cells (*Figure 2D*). This suppression, however, was partially reversed by mutating individual CpG motifs, and was completely absent with a CpG-null mutant of *CRP* promoter. These results suggest that promoter methylation inhibits, whereas its demethylation enhances *CRP* expression, thus supporting a causal association.

CpG methylation and demethylation are mediated by DNA methyltransferases (DNMTs) and ten-eleven translocations (TETs), respectively (*Jones, 2012*; *Wu and Zhang, 2014*; *Dor and Cedar, 2018*; *Luo et al., 2018*). A causal association with promoter methylation would therefore predict that the expression of *CRP* should also be regulated by DNMTs and/or TETs. Indeed, RNAi screening revealed that knockdown (KD) of *DNMT3A* enhanced (*Figure 3A*), whereas KD of *TET2* reduced *CRP* expression in Hep3B cells (*Figure 3B*). Knockout (KO) of *DNMT3A* (*Figure 3C*) or *TET2* (*Figure 3D*) with Cas9 yielded consistent but more pronounced effects. Importantly, *DNMT3A/TET2* KD or KO showed expected effects on *CRP* promoter methylation and on their own expression (*Figure 3—figure supplements 1* and *2*) without upregulating TFs critical to *CRP* induction (*Figure 3—figure supplement 3*). On the other hand, the overexpression of *DNMT3A* reduced, while the overexpression of *TET2* enhanced *CRP* expression (*Figure 3E*). These results together identify DNMT3A and TET2 as the negative and positive regulators of *CRP* expression, respectively, thus reinforcing the notion that *CRP* expression is causally determined by promoter methylation.

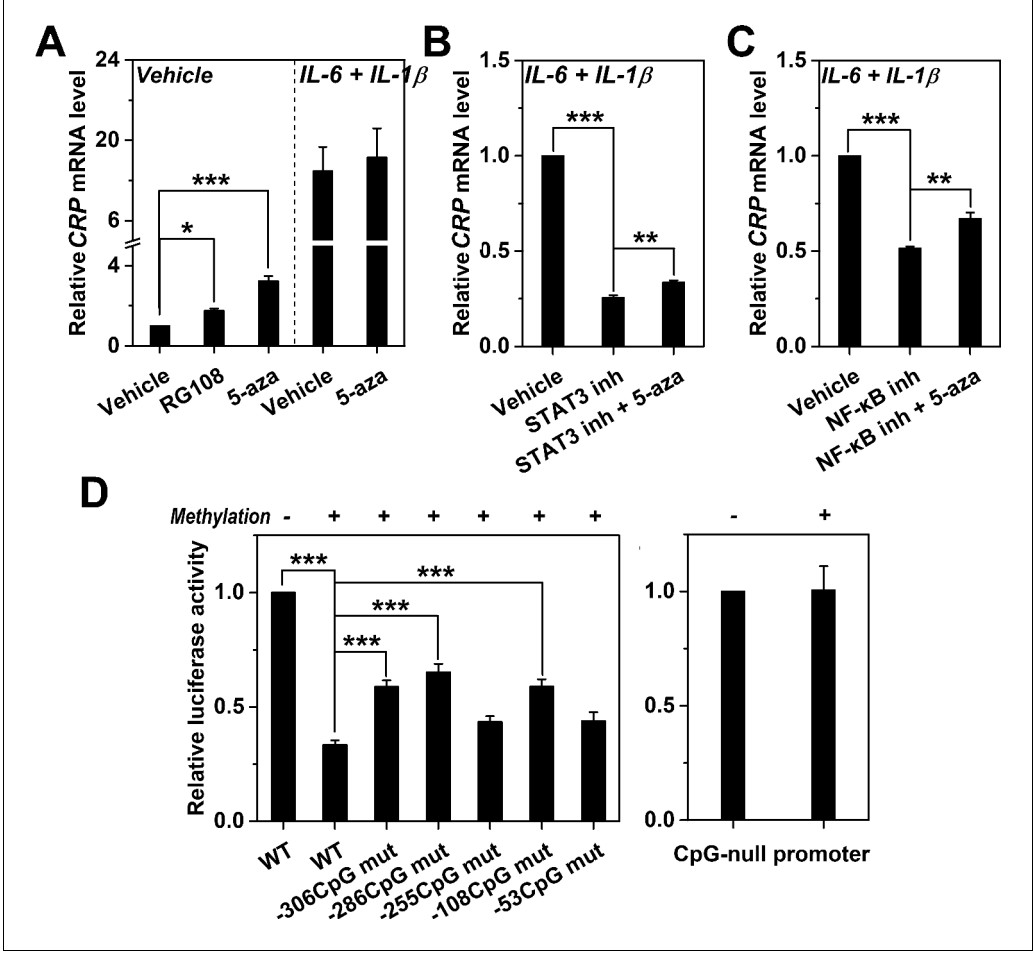

**Figure 2.** Methylation levels of *CRP* promoter causally determines expression. (**A**) The effects of DNA methylation inhibitor RG-108 (25 μM, 24 hr) or 5-aza (5 μM, 12 hr) on *CRP* expression in Hep3B cells at the resting or induced state (n = 3). These inhibitors enhanced the resting but not the induced expression of *CRP*. At the induced state, the defective *CRP* expression caused by STAT3 (s31-201, 30 μM, 24 hr) (**B**) or NF-κB inhibition (BAY11-7082, 2 μM, 24 hr) (**C**) was partially reversed by 5-aza (5 μM, 24 hr) (n = 3). (**D**) In vitro vector methylation markedly inhibited reporter activities of wildtype *CRP* promoter (WT) following transfection into Hep3B cells (n = 3). Mutating individual CpG motif partially reversed this inhibition. As the control, in vitro vector methylation did not affect reporter activities of a CpG-null version of *CRP* promoter. Data are presented as mean ± SEM. *p<0.05, **p<0.01, ***p<0.001 (two-tailed t-test).

The online version of this article includes the following figure supplement(s) for figure 2:

**Figure supplement 1.** Validation of effects on promoter methylation of *CRP* by DNA methylation inhibitor.

Of note, inhibitor treatment or *DNMT3A/TET2* manipulations would all affect the methylation status of entire genome. To exclude any indirect effect caused by global manipulation, we specifically modulated the methylation levels of *CRP* promoter by dCas9-mediated targeting of the catalytic domains of DNMT3A or TET2. Enforced methylation of *CRP* promoter by DNMT3A-dCas9 reduced the expression of *CRP* in Hep3B cells, but showed little effect on that of *serum amyloid A* (*SAA*, another major human APR) and *serum amyloid P component* (*SAP*, a paralog of *CRP*) (*Figure 3F*). By contrast, enforced demethylation of *CRP* promoter by TET2-dCas9 only selectively enhanced the expression of *CRP* (*Figure 3G*). We thus conclude that DNMT3A and TET2-tuned methylation status of *CRP* promoter constitutes a key part of the regulatory mechanism that causally determines the expression.

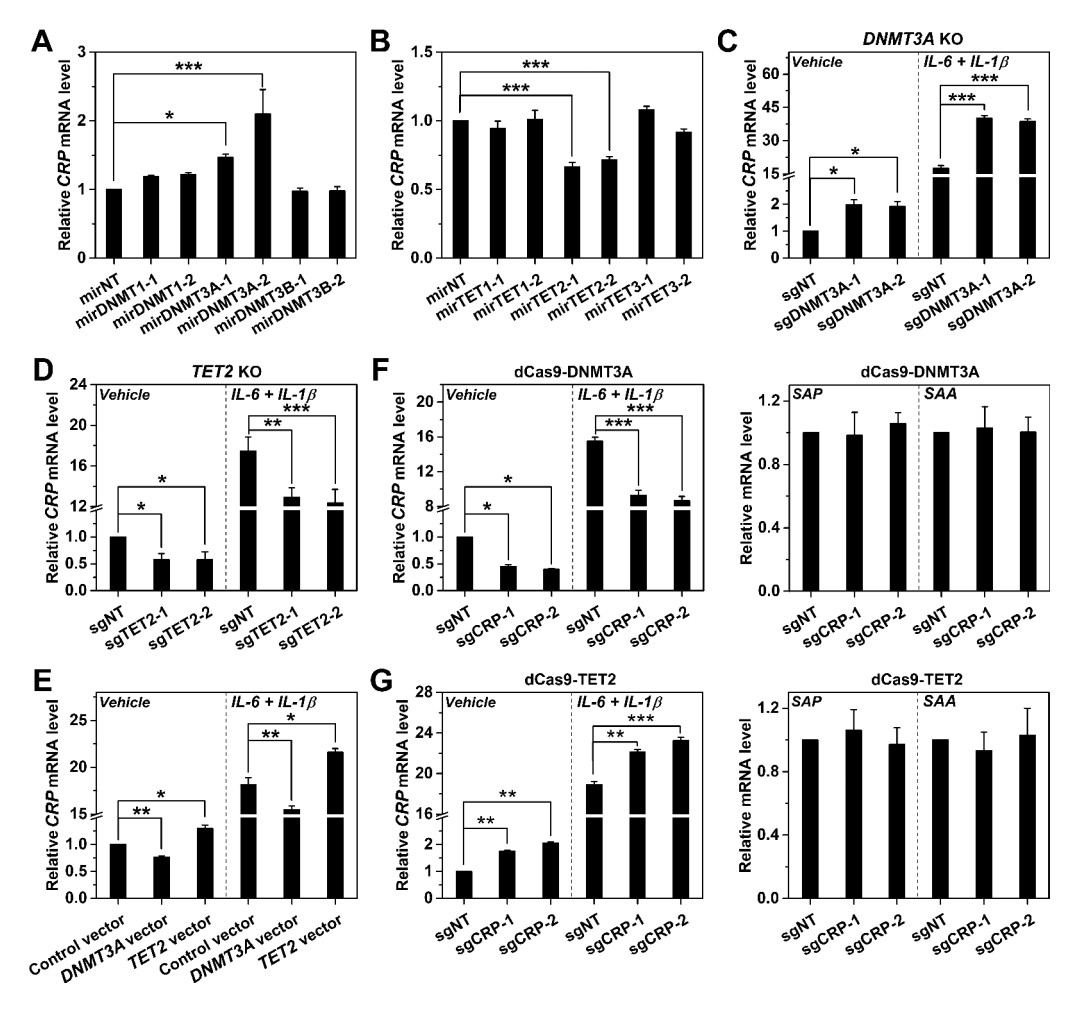

**Figure 3.** DNMT3A and TET2 regulate *CRP* expression. *CRP* expression in Hep3B cells with *DNMT* (A) or *TET* (B) knockdown by control (mirNT) or target-specific miRNA (n = 3). (C) *CRP* expression in Hep3B cells with co-transfected Cas9 and sgRNAs targeting exon 14 (sgDNMT3A-1) or 2 (sgDNMT3A-2) of *DNMT3A* (n = 3). (D) *CRP* expression in Hep3B cells with co-transfected Cas9 and sgRNAs targeting exon 3 (sgTET2-1) or 7 (sgTET2-2) of *TET2* (n = 3). (E) *CRP* expression in Hep3B cells with overexpressed *DNMT3A* or *TET2* (n = 3). *CRP* expression in Hep3B cells with co-transfected catalytic domain of *DNMT3A* (F) or *TET2* (G) fused to *dCas9* and sgRNAs targeting *CRP* promoter (n = 3). The results identified DNMT3A and TET2 as the negative and positive regulators of *CRP* expression, respectively. Selective targeting of DNMT3A or TET2 to *CRP* promoter by dCas9 only regulated the expression of *CRP*, but did not affect that of *serum amyloid A* (*SAA*; a major human APR) or *serum amyloid P component* (*SAP*, a paralog of *CRP*). Data are presented as mean ± SEM. *p<0.05, **p<0.01, ***p<0.001 (two-tailed t-test).

The online version of this article includes the following figure supplement(s) for figure 3:

**Figure supplement 1.** Validation of effects on promoter methylation of *CRP* by DNA methylation-modulating manipulations.

**Figure supplement 2.** Validation of *DNMT3A* and *TET2* knockdown and knockout.

**Figure supplement 3.** Effects on TF expression by DNA methylation-modulating manipulations.

## Promoter methylation of CRP dictates strength of TF recruitment

We next asked whether the promoter methylation-mediated regulation could be conferred by influencing TF recruitment. Indeed, IL-6 and IL-1β-induced demethylation of *CRP* promoter (*Figure 1G*) was paralleled by markedly enhanced recruitment of STAT3, NF-κB p50 and C/EBP-β (*Figure 4A*; *Singh et al., 2007*; *Young et al., 2008*). Moreover, in vitro methylation substantially reduced the recruitment of those TFs to vectors containing *CRP* promoter after transfection into Hep3B cells (*Figure 4B*). The −53CpG and −108CpG are at the binding sites of p50/C/EBP-β, and STAT3, respectively (*Figure 1A*). Accordingly, site-specific methylation of −53CpG selectively prevented the recruitment of p50 and C/EBP-β to *CRP* promoter, while site-specific methylation of −108CpG only

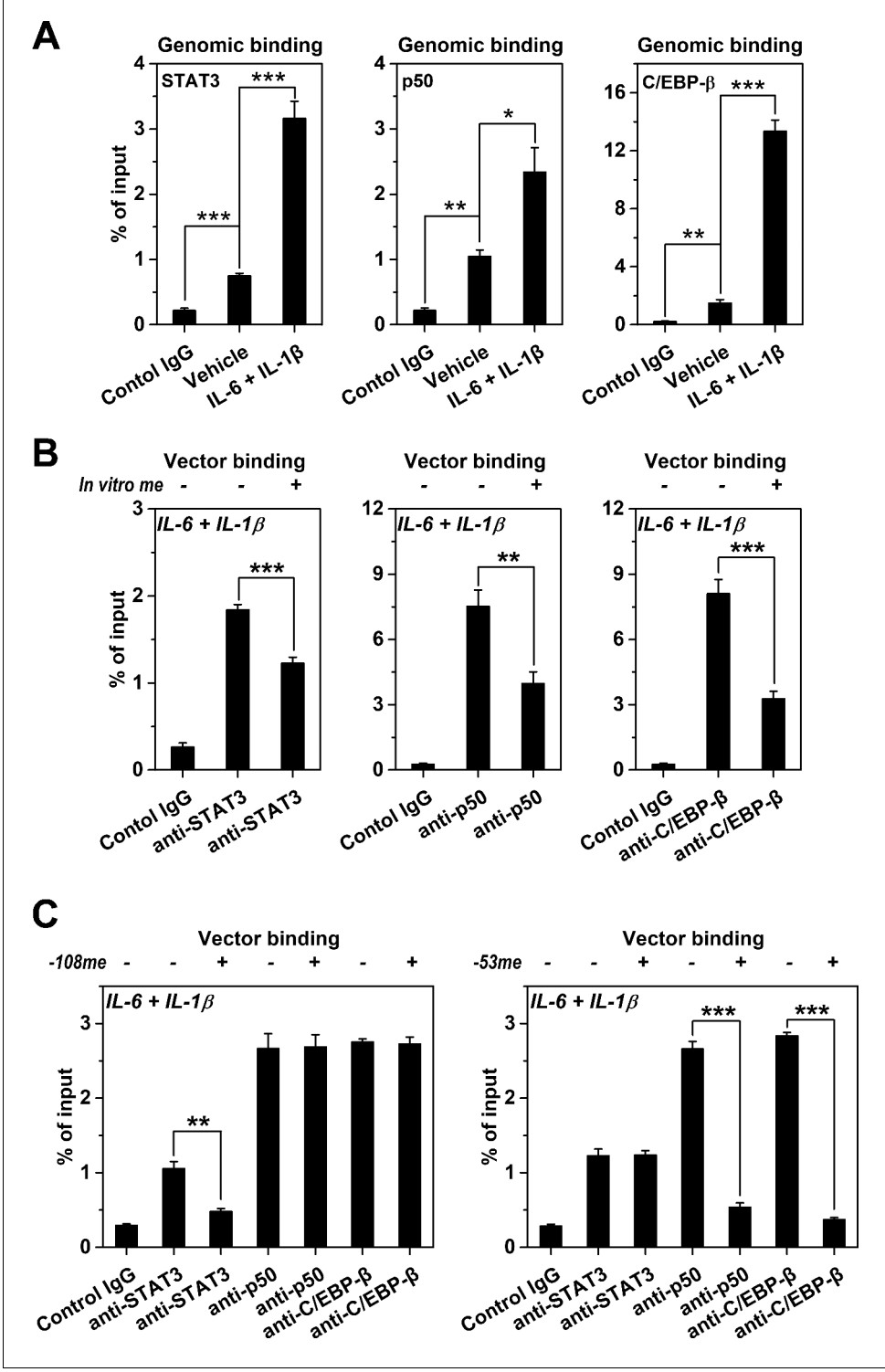

**Figure 4.** Methylation levels of *CRP* promoter affect TF recruitment. The recruitment of TFs to *CRP* promoter in Hep3B cells was analyzed by ChIP. (**A**) The recruitment of STAT3, p50 and C/EBP-β to *CRP* promoter were all markedly enhanced at the induced versus resting state (n = 3). (**B**) In vitro vector methylation decreased the recruitment of STAT3, p50 and C/EBP-β to a vector containing *CRP* promoter at the induced state (n = 3). (**C**) Site-specific methylation at −53CpG inhibited the recruitment of p50 and C/EBP-β, whereas methylation at −108CpG inhibited the recruitment STAT3 to the vector containing *CRP* promoter at the induced state (n = 3). Data are presented as mean ± SEM. *p<0.05, **p<0.01, ***p<0.001 (two-tailed t-test).

inhibited the recruitment of STAT3 (*Figure 4C*). These CpG motifs at TF binding sites may thus act as rheostats with their methylation turning down the recruitment of critical TFs, resulting in reduced expression.

## Dynamic crosstalk among TFs and promoter methylation in induced expression of CRP

We further examined how TF recruitment and promoter methylation dynamically orchestrate to regulate the induced expression of *CRP*. IL-6 and IL-1β induced two waves of *CRP* expression: the first wave lasted from 0 to 6 hr yielding the minor peak, while the second lasted from 12 to 24 hr yielding the major peak (*Figure 5A*). The recruitment of STAT3 occurred during the first wave and saturated at 3 hr before the minor peak (*Figure 5B*). By contrast, the recruitment of p50 was more evident during the time lag between the two waves (*Figure 5C*). The recruitment of C/EBP-β, whose action depends on p50 (*Cha-Molstad et al., 2000*; *Kramer et al., 2008*; *Agrawal et al., 2001*), however, steadily rose till 12 hr (*Figure 5D*). These would suggest that the first wave of induced *CRP* expression is driven by early recruited STAT3, which licenses the late recruitment of p50 that synergizes with C/EBP-β to drive the second wave. As such, STAT3 is likely the pioneer TF that binds methylated *CRP* promoter to initiates induction and primes demethylation.

In line with the above suggestion, STAT3 was the only TF showing appreciable early recruitment to *CRP* promoter upon enforced DNA methylation by *TET2* KO (*Figure 5B–D*). This indicates that STAT3 can nevertheless be recruited to *CRP* promoter even when heavily methylated, consistent with the observations that vector binding of STAT3 was least sensitive to methylation (*Figure 4B and C*). This also indicates that STAT3 can act largely independent of p50 and C/EBP-β to drive *CRP* induction, albeit with a markedly reduced amplitude (*Figure 5A*). Indeed, the sole activation of STAT3 in wildtype cells was able to induce *CRP* expression to a level comparable to that of the minor peak, whereas the sole activation of NF-κB was completely ineffective (*Figure 5E*). Despite that, NF-κB inhibition (with intact STAT3) at the induced state resulted in an even stronger methylation of *CRP* promoter (*Figure 5F*) and a reversal of allelic imbalance (*Figures 5G* and *1F*). Therefore, promoter demethylation requires p50 that acts downstream of STAT3.

Interestingly, enforced DNA demethylation by *DNMT3A* KO not only tripled *CRP* expression during the entire course of induction, but eliminated the time lag between the two waves (*Figure 5A*). The augmented amplitude can be explained by the enhanced recruitment of the three TFs, while the altered dynamics may correspond to the shifted timing of p50 recruitment (*Figure 5B–D*). As such, p50 selectively recruited during the time lag could be responsible for promoter demethylation to prime the second wave. Accordingly, C/EBP-β appears to be the major effector that responds to promoter demethylation: its overexpression did not demethylate *CRP* promoter (*Figure 5H and I*), but when combined with blockage of DNA methylation, it drove the resting expression of *CRP* to a level approaching to that induced by IL-6 and IL-1β (*Figure 5J*). C/EBP-β KO, however, lowered the induced expression of *CRP* by ~70% (*Figure 5K*). These together demonstrate a stepwise induction of *CRP* where TFs and promoter methylation dynamically orchestrate (*Figure 5L*).

## Reversible methylation regulates expression of genes with CpG-poor promoters

Having established the regulation of *CRP* expression by reversible promoter methylation, we wondered whether the same regulation can be applied to other APRs. Indeed, *SAA* behaved similarly as *CRP*. Treating Hep3B cells with IL-6 and IL-1β resulted in a drastic increase in the expression of *SAA* (*Figure 6A*) and a reduction in methylation levels of its promoter (*Figure 6B*). These were, however, quickly recovered following cytokine withdrawal. By contrast, neither the expression nor the promoter methylation of *SAP* was affected by treatment or withdraw of IL-6 and IL-1β (*Figure 6A and C*). Moreover, *DNMT3A* KO also markedly enhanced the induced expression of *SAA*, but barely affected that of *SAP* (*Figure 6D*). These results suggest that reversible promoter methylation may be a general mechanism underlies the induction of APRs.

The promoters of most mammalian genes contain a high frequency (observed number/expected number >0.6) of CpGs termed CpG islands (CGIs) that are resistant to DNA methylation (*Saxonov et al., 2006*). The CpG frequency of CRP promoter, however, is exceptionally low (~0.23). Interestingly, a low CpG frequency appears to be general feature of APR promoters (*Figure 7A*).

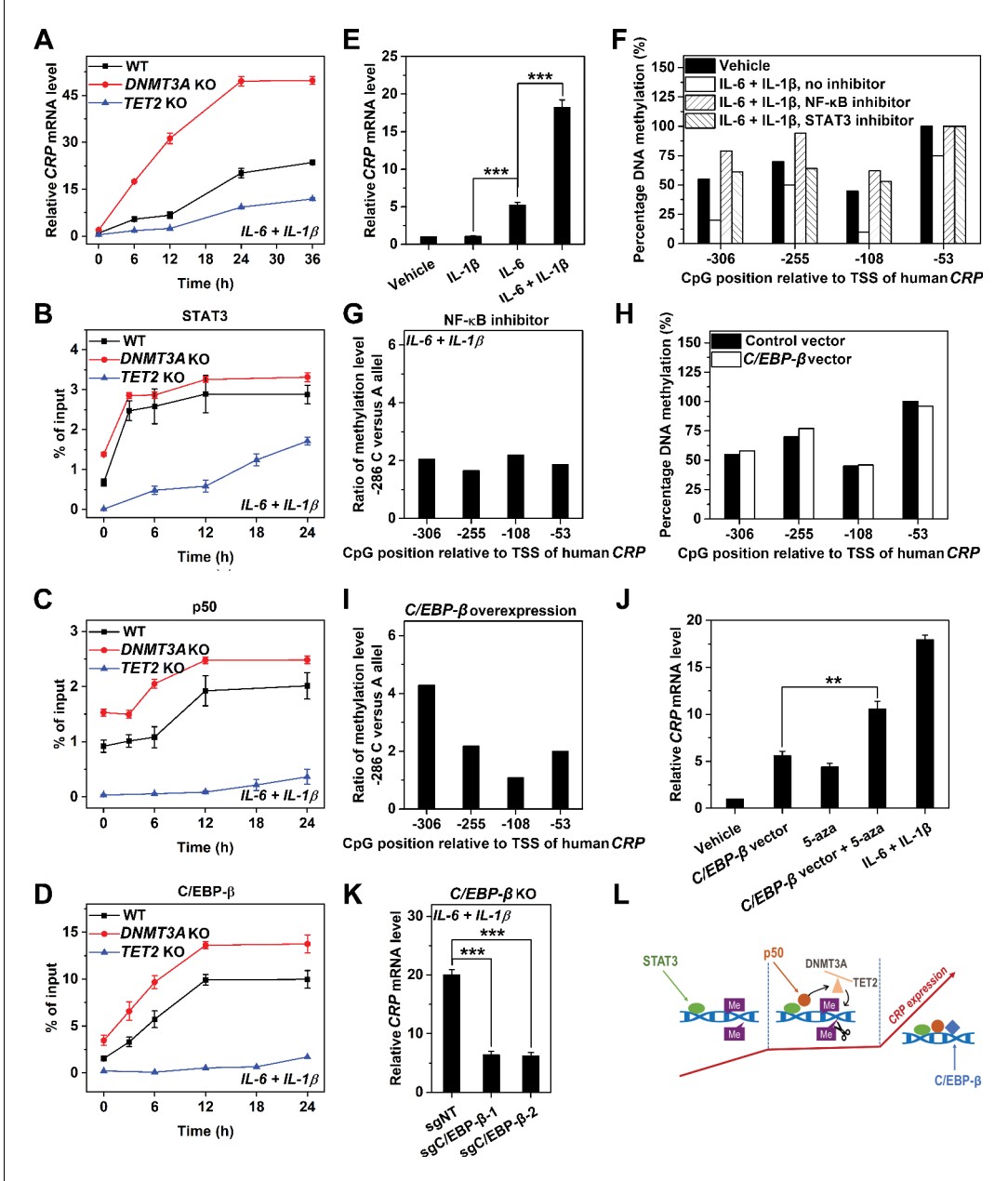

**Figure 5.** TF recruitment and promoter methylation dynamically crosstalk to regulate the induced expression of *CRP*. *CRP* expression (**A**) and promoter recruitment of STAT3 (**B**), p50 (**C**) and C/EBP-β (**D**) in Hep3B cells at the induced state without (WT) or with co-transfected *Cas9* and sgRNA targeting exon 14 of *DNMT3A* (n = 3) or targeting exon 3 of *TET2* (n = 2) over time. *TET2* KO markedly reduced the induction of *CRP*, and almost abrogated the promoter recruitment of p50 and C/EBP-β. The recruitment of STAT3 to *CRP* promoter was still evident in *TET2* KO cells. *DNMT3A* KO resulted in a stronger amplitude and altered dynamics of *CRP* induction. The recruitment to *CRP* promoter was enhanced for all the three TFs in *DNMT3A* KO cells, whereas the timing of recruitment was altered only for p50. (**E**) *CRP* expression in Hep3B cells treated with vehicle, 1 ng/ml IL-1β, 10 ng/ml IL-6 or their combination for 48 hr (n = 3). As IL-1β is unable to induce IL-6 production in Hep3B cells (*Kramer et al., 2008*), the effects of STAT3 and p50 can be largely dissociated by treating cells with one single cytokine (*Kramer et al., 2008*; *Ganapathi et al., 1991*; *Ganapathi et al., 1988*). IL-1β could not induce *CRP* expression, suggesting p50 is not required for the first wave of *CRP* induction. (**F**) Methylation levels of *CRP* promoter in Hep3B cells at the induced state treated without (Vehicle) or with inhibitors of STAT3 (30 μM s31-201) or NF-κB (2 μM BAY11-7082) for 24 hr. (**G**) Ratios of methylation levels on −286C versus −286A alleles in Hep3B cells at the induced state treated with the NF-κB inhibitor (2 μM BAY11-7082) for 24 hr. Methylation levels (**H**) and allelic methylation of *CRP* promoter (**I**) in Hep3B cells expressing a control or a *C/EBP-β* vector at the resting state. *C/EBP-β* overexpression showed no effect on methylation status of *CRP* promoter. The result of one representative experiment is shown. (**J**) *CRP* expression in Hep3B cells with or without *C/EBP-β* overexpression under the indicated conditions for 48 hr (n = 3). (**K**) *CRP* expression in Hep3B cells without (sgNT) or with *C/EBP-β* KO (sgC/EBP-β) following induction with 10 ng/ml IL-6 and 1 ng/ml IL-1β for 48 hr (n = 3). The dramatic effects of *C/EBP-β* KO or

*Figure 5 continued on next page*

*Figure 5 continued*

overexpression suggest that this TF is the major effector that respond to promoter methylation status and determine the amplitude of *CRP* expression. (L) A schematic illustration of how TF recruitment and promoter methylation dynamically orchestrate to regulate the induction of *CRP*. Data are presented as mean ± SEM. **p<0.01, ***p<0.001 (two-tailed t-test).

We then extended our analysis to genes with CpG-poor promoters. In Hep3B cells treated with IL-6 and IL-1β, strongly induced genes tended to manifest lower CpG ratios in their promoters

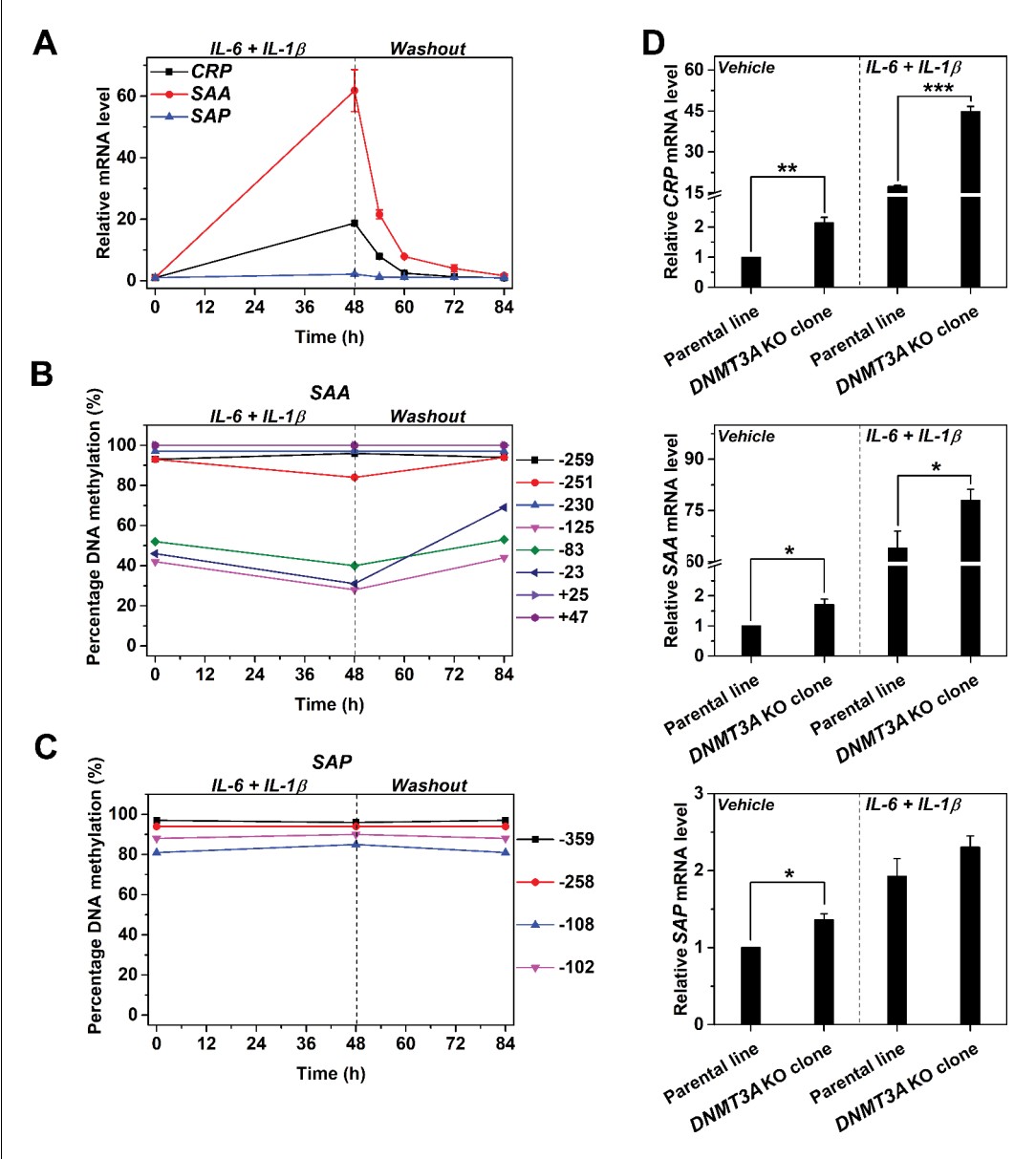

**Figure 6.** Reversible methylation regulates the induced expression of *SAA*. Hep3B cells were treated with 10 ng/ml IL-6 and 1 ng/ml IL-1β for 48 hr, and then cultured in the absence of cytokine for 36 hr. (A) The expression levels of *CRP, SAA* and *SAP* (n = 3). (B) The methylation levels of *SAA* promoter. (C) The methylation levels of *SAP* promoter. The expression and methylation levels of *SAA* were inversely, and dynamically coupled. The expression of *SAP* was not induced, and methylation levels of its promoter did not change over time. The result of one representative experiment is shown. (D) The expression levels of *CRP* (upper), *SAA* (middle) and *SAP* (lower) in the parental or a clone of *DNMT3A* KO Hep3B cells at the resting or induced state (n = 3). *DNMT3A* KO enhanced the expression of *CRP* and *SAA*, but showed only marginal effects on that of *SAP*. Data are presented as mean ± SEM. *p<0.05, **p<0.01, ***p<0.001 (two-tailed t-test).

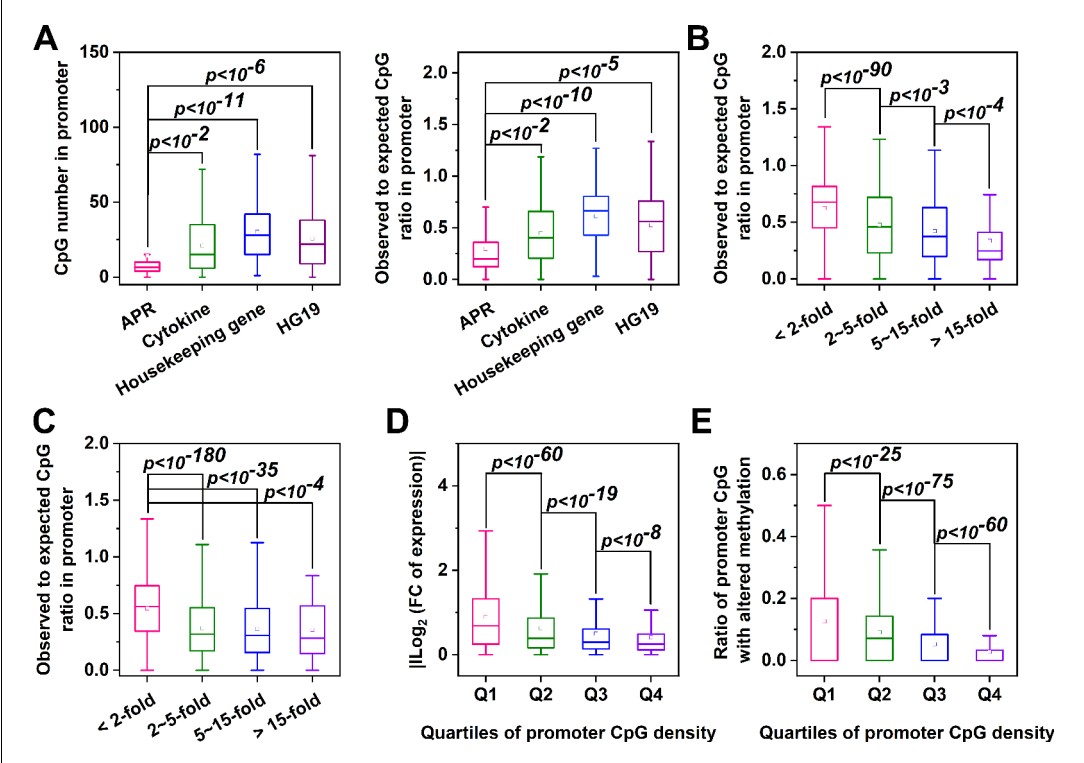

**Figure 7.** Genes with CpG-poor promoters are preferentialy demethylated and induced in acute inflammation. (A) Both the numbers and ratios of promoter CpG are significantly lower in APRs (n = 34) than in cytokines (n = 117), housekeeping genes (n = 425) or all genes of HG19 (n = 20180). (B) Hep3B cells were treated with 10 ng/ml IL-6 and 1 ng/ml IL-1β for 24 hr, and gene expression profiles were determined by RNA-seq. Genes with larger expression changes exhibited lower promoter CpG ratios. There are 10979, 2130, 454, and 121 genes in <2 fold, 2 ~ 5 fold, 5 ~ 15 fold, and >15 fold categories, respectively. Mouse liver tissues were collected at the resting or turpentine-induced state. Their transcriptome and methylome were then determined and compared. (C) Genes whose expression changed by over 2-fold between the two states exhibit lower promoter CpG densities. There are 12338, 2876, 444, and 51 genes in <2 fold, 2 ~ 5 fold, 5 ~ 15 fold, and >15 fold categories, respectively. With the increase in promoter CpG density, genes show markedly reduced changes in levels of their expression (D) (There are 3096, 3903, 3903, and 3907 genes in Q1, Q2, Q3, and Q4 categories, respectively) and promoter methylation (E) (There are 3745, 3738, 3741, and 3742 genes in Q1, Q2, Q3, and Q4 categories, respectively). Statistical analysis was performed using K-S test.

The online version of this article includes the following figure supplement(s) for figure 7:

**Figure supplement 1.** Correlation and GO analysis.

(*Figure 7B*; *Figure 7—figure supplement 1*). Similar observations were also made in livers of mice undergoing turpentine-induced acute inflammation (*Figure 7C*). Importantly, genes with CpG-poor promoters showed significantly stronger changes in both expression and promoter methylation (*Figure 7D and E*; *Figure 7—figure supplement 1*). Therefore, dynamic methylation may also regulate the expression of highly inducible genes with CpG-poor promoters.

## Discussion

Though TFs critical to *CRP* expression have been identified, how their actions are coordinated remains unclear. This study demonstrates a previously unrecognized, epigenetic mechanism wherein methylation status of *CRP* promoter responds to and further modifies the effects of distinct TFs. At the induced state, the pioneered binding of STAT3 to *CRP* promoter in hepatocytes drives a minor wave of induction, and further licenses the subsequent recruitment of NF-κB p50. The two TFs probably work together to tip the balance of TET2 and DNMT3A at *CRP* promoter, leading to its demethylation. As significant cell proliferation was not noted, the methylated cytosine might be eventually removed by base excision repair mechanism (*Wu and Zhang, 2014*). Consequently, the demethylated promoter enhances the recruitment of C/EBP-β to drive the major wave of *CRP*

induction. During the recovery phase, however, the loss of activated STAT3 and p50 results in a rapid remethylation of *CRP* promoter and termination of induction. At the resting state, however, the relatively hypomethylated promoter of *CRP* in the liver versus other tissues likely favors C/EBP-β recruitment, contributing to its tissue-specific, basal expression. These may form the basis for CRP, a putative pattern recognition receptor (*Du Clos, 2013*; *Bottazzi et al., 2010*), to constitute an integral part of immune surveillance in both homeostasis and inflammation.

In addition to *CRP*, we further show that reversible methylation also appears to be involved in regulation of highly inducible genes carrying CpG-poor promoters with APRs as representatives. In this regard, it is worth noting that DNA methylation is a relative stable epigenetic modification (*Wu and Zhang, 2014*). Though dynamic changes in global or local DNA methylation status have been demonstrated in processes of development, aging and disease, these (gradual) changes, once occurred, are largely persistent or irreversible (*Wu and Zhang, 2014*; *Dor and Cedar, 2018*; *Luo et al., 2018*; *Halder et al., 2016*; *Sellars et al., 2015*; *Domcke et al., 2015*; *Flavahan et al., 2016*; *Dmitrijeva et al., 2018*; *Koch et al., 2018*; *Horvath and Raj, 2018*) with its causality in determining gene expression even being questioned (*Bestor et al., 2015*). There are, however, only rare cases where the expression of specific genes, such as *pS2* (*Métivier et al., 2008*; *Kangaspeska et al., 2008*) and *IL-10* (*Hwang et al., 2018*), is regulated by rapid and reversible changes in DNA methylation. Our findings here identify an important scenario in which reversible promoter methylation plays a critical role in determining the expression pattern of a class of proteins featured by CpG-poor promoters in response to inflammatory stimuli.

Acute changes in DNA methylation have also been analyzed at a genome-wide scale in previous studies. One study examined mouse neurons activated by electroconvulsive stimulation, and found that methylation changes were preferentially occurred in CpG-poor regions (*Guo et al., 2011*). Though promoters were underrepresented in these regions, their methylation changes were nevertheless modestly anticorrelated with gene expression (*Guo et al., 2011*). Such CpG content-dependent changes in DNA methylation, however, were not observed in another study examining neurons activated by contextual learning (*Halder et al., 2016*). Moreover, DNA methylation changes in human dendritic cells following infection rarely occurred at promoters (*Pacis et al., 2019*; *Pacis et al., 2015*), and were claimed to be a consequence of gene expression (*Pacis et al., 2019*). Those findings thus argue that the regulation of reversible methylation on inducible expression of genes with CpG-poor promoters may be context-dependent, and that rigorously controlled case study should be integrated into genome-wide investigation to conclude on causality.

Interestingly, genes showing inducible expression in macrophages activated by endotoxin have been classified into two groups: nucleosome remodeling-independent genes with CpG-rich promoters, and nucleosome remodeling-dependent genes with CpG-poor promoters (*Ramirez-Carrozzi et al., 2009*). Despite their distinct requirements for SWI/SNF complexes, preassembled Pol II and new protein synthesis, both groups of genes exhibit a single-wave kinetics of induction (*Ramirez-Carrozzi et al., 2009*; *Hargreaves et al., 2009*; *Ramirez-Carrozzi et al., 2006*). This is in contrast with CRP and probably other major APRs, which show a two-wave kinetics of induction with the second wave licensed by promoter demethylation. However, nucleosome remodeling might still act downstream in the second wave, as C/EBP-β has recently been shown to promote CRP expression by recruiting BRG1 via MKL1 (*Fan et al., 2019*). Conversely, promoter methylation could also contribute to inducible expression of nucleosome remodeling-dependent genes in macrophages by directly regulating TF recruitment (*Hu et al., 2013*; *Yin et al., 2017*; *Thomas et al., 2012*).

Despite that most CpGs in mammalian promoters not associated with CGIs are usually methylated (*Wu and Zhang, 2014*; *Luo et al., 2018*), TETs and DNMTs can nevertheless mediate active demethylation and de novo methylation, respectively. This indicates that the methylation status of part of the genome may depend on the balance of the two types of enzymes (*Jones, 2012*; *Wu and Zhang, 2014*; *Dor and Cedar, 2018*; *Luo et al., 2018*; *Blattler and Farnham, 2013*). Cellular signaling able to tip the balance of TETs and DNMTs could thus represent a regulatory mechanism that finely and reversibly tunes the expression of certain genes in response to environmental cues. In case of *CRP*, DNMT3A and TET2 appear to be involved in the regulation. Though the mechanism of TF-induced local demethylation is not fully understood (*Luo et al., 2018*), TFs including NF-κB and EGR1 have been reported to evoke DNA demethylation in neurons (*Jarome et al., 2015*) for example by recruiting TET1 (*Sun et al., 2019*). Future study is warranted to elucidate how TFs regulate

the balance of TETs and DNMTs and to discover scenarios where regulation by reversible DNA methylation plays a prominent role.

## Materials and methods

### Determination of promoter methylation level

Frozen tumor/normal tissue sample pairs were obtained from the tissue bank of Gansu Cancer hospital. Genomic DNA samples extracted from tissues or Hep3B cells were bisulfite-converted and recovered using EpiTect Bisulfite Kit (QIAGEN, Hamburg, Germany; catalog number: 59104; lot number: 142338839, 145038568, 148214306, 151030901) according to the manufacturer's instructions. Samples were then amplified with SureStart Taq DNA Polymerase (Agilent Technologies, Santa Clara, CA; catalog number: 600282; lot number: 0006129848). The primer sequences used were: human *CRP* (Forward: 5'-GTAGGTGTTGGAGAGGTAGTTATTA-3'; Reverse: 5'-ATTTATA TCCAAAACAATAAAAAAATTTAC-3'); rabbit *CRP* (Forward: 5'-ATGTTAGAGTTGAAGGTG TTGGAGATA-3'; Reverse: 5'-AAATACTAAAAATCCTACATCCCTTACCTC-3'); human *SAA* (Forward: 5'-GTTTTTATTTTATATTTTTTAGTAG-3'; Reverse: 5'-TAATACTAATCTATACTATAACTAAACTAC-3'); human *SAP* (Forward: 5'-AAGAAAGAAAAGGTTTTGTTTTTA-3'; Reverse: 5'-ATTTTCCAAATC TACCTCCTAAC-3')). Subsequent cloning and sequencing were performed as described (*Varley et al., 2009*). The experiments conformed to the Guide for the Care and Use of Laboratory Animals published by NIH, and were conducted according to the protocols approved by the Ethics Committee of Animal Experiments of Xi'an Jiaotong University and Lanzhou University (2016–064 and A201307050027).

### Determination of gene expression

Human hepatocellular carcinoma cell line Hep3B were obtained from cell bank of Chinese Academy of Sciences (Shanghai, China). Hep3B cells were cultured in MEM media (Sigma-Aldrich, St. Louis, MO; catalog number: M0643; lot number: SLBF6418, SLBM7544V) containing 10% FBS (Biological industries, Beit Haemek, Israel; catalog number: 04-001-1A; lot number: 1418110). Cells were tested for mycoplasma and have been authenticated by Cel-ID (*Mohammad et al., 2019*) using RNA-seq: their correlation ($R^2$) to Hep3B cells (G28888.Hep_3B2.1.7.3) in Cancer Cell Line Encyclopedia project is 0.92 ($p<10^{-24}$). Total RNA was extracted with RNAiso Plus reagent (Takara, Shiga, Japan; catalog number: 9108, 9109; lot number: 2270A, AA4102-1, AA6201-1, AA3101-1). cDNA was synthesized from 1 μg total RNA using reverse transcriptase M-MLV (Takara; catalog number: 2641B; lot number: AG70412A), Oligo d(T)15 Primers (Takara; catalog number: 3805; lot number: T1301BA), dNTP mixture (Takara; catalog number: 4019; lot number: B4101A), and recombinant ribonuclease inhibitor (Takara; catalog number: 2313B; lot number: K8101GA). Gene expression was determined with quantitative PCR (q-PCR) using SYBR Premix Ex Taq II (Takara; catalog number: RR820A; lot number: AK7602) in a CFX96 Real-Time PCR Detection System (Bio-rad, Hercules, CA). The gene expression levels were normalized to that of ACTB. The primer sequences used were: human *CRP* (Forward: 5'-GGAGCAGGATTCCTTCGGT-3'; Reverse: 5'-CACTTCGCCTTGCACTTCAT-3'); human *SAA* (Forward: 5'-GTGATCAGCGATGCCAGAGAGA-3'; Reverse: 5'-CCAGCAGG TCGGAAGTGATTG-3'); human *SAP* (Forward: 5'-CTTGATCACACCGCTGGAGAAG-3'; Reverse: 5'-CTTGGGTATTGTAGGAGAAGAGGCTG-3'); human *ACTB* (Forward: 5'-CGTGGACATCCGCAAA-GAC-3'; Reverse: 5'- CTCAGGAGGAGCAATGATCTTGA-3').

Gene expression profiles of Hep3B cells with or without IL-6 and IL-1β treatment for 24 hr were determined with RNA sequencing services provided by GENEWIZ (South Plainfield, NJ). Transcriptome and methylome of C57BL/6 mouse liver tissues at the resting or turpentine-induced state were determined with RNA sequencing and whole genome bisulfite sequencing services (WGBS) provided by GENEWIZ. All RNA-seq and WGBS data have been deposited in GEO under accession code GSE146797.

Where appropriate, Hep3B cells were treated with 10 ng/ml rhIL-6 (R and D Systems, Minneapolis, MN; catalog number: 206-IL; lot number: OJZ1716061), 1 ng/ml rhIL-1β (R and D System; catalog number: 201-LB; lot number: AD1515091), 5-Azacytidine (5-aza; 5 μM, 12 hr; DNA methylation inhibitor) (Sigma-Aldrich; catalog number: A2385; lot number: SLBL4994V), RG108 (25 μM, 24 hr; DNA methylation inhibitor) (Selleck Chemicals, Houston, TX; catalog number: S2821; lot number: 02),

BAY11-7082 (2 μM, 24 hr; NF-κB inhibitor) (Selleck Chemicals; catalog number: S2913; lot number: 01), S31-201 (30 μM, 24 hr; STAT3 inhibitor) (Selleck Chemicals; catalog number: S1155; lot number: 02) or Stattic (5 μM, 24 hr; STAT3 inhibitor) (Selleck Chemicals; catalog number: S7024; lot number: 01).

## Modulation of DNA methylation regulators

For knockdown, targeting sequences were synthesized (GENEWIZ), and cloned into pcDNA6.2-GW/ EmGFP-miR vector (Invitrogen, Carlsbad, CA; catalog number: K4935-00). The targeting sequences were: *DNMT1* (#1 5'-GATTTGGAAAGAGACAGCTTA-3'; #2 5'-CAACAGAGGACAACAAGTTCA-3'); *DNMT3A* (#1 5'-GGTGTGTGTTGAGAAGCTGAT-3'; #2 5'-GAATTTGACCCTCCAAAGGTT-3'); *DNMT3B* (#1 5'-GGTTTGGCGATGGCAAGTTCT-3'; #2 5'-CGAGAACAAATGGCTTCAGAT-3'); *TET1* (#1 5'-CATGCAAGGCCTTCCAGATTA-3'; #2 5'-AGAGAACAGCCAGTTTGCTTA-3'); *TET2* (#1 5'-GTGTAGGTAAGTGCCAGAAAT-3'; #2 5'-CATGGCGTTTATCCAGAATTA-3'); *TET3* (#1 5'-CCTTTATGACTTCCCTCAGCG-3'; #2 5'-CCAGTTGATGGACCTGTTCCA-3'). For overexpression, coding sequence of target genes were cloned into pcDNA3.1 vector (Invitrogen; catalog number: V795-20). Vectors were transfected into Hep3B cells with ViaFect Transfection Reagent (Promega, Madison, WI; catalog number: E4982; lot number: 0000251076, 0000136819). 48 hr later, gene expression was determined with q-PCR.

For Cas9-mediated knockout (*Ran et al., 2013*), targeting sequences corresponding to sgRNAs were synthesized (GENEWIZ), and cloned into pSpCas9(BB)−2A-Puro (PX459) v2.0 vector (Addgene, Cambridge, MA; catalog number: 62988). The targeting sequences were: *DNMT3A* (#1 5'-GGACCTCTTGGTGGGGCCGG-3' against exon 14; #2 5'-GGAAGGTGGGGCGGCCTGGG-3' against exon 2); *TET2* (#1 5'-GGGAGATGTGAACTCTGGGA-3' against exon 3; #2 5'-GGAGAACTTGCGCCTGTCAG-3' against exon 7). Gene expression was determined with q-PCR. In some experiments, single clones of *DNMT3A* knockout cells were further selected with puromycin (Corning, NY; catalog number: 58-58-2; lot number: 61385051).

For dCas9-mediated targeting (*Hilton et al., 2015*; *Kabadi et al., 2014*), targeting sequences corresponding to sgRNAs were synthesized (GENEWIZ), and cloned into phU6-gRNA vector (Addgene; catalog number: 53188). The targeting sequences for *CRP* promoter were: #1 5'-GGGGACTGTTGTGGGGTGGG-3'; #2 5'-GAAGCTCTGACACCTGCCCC-3'. The catalytic domains of *DNMT3A* and *TET2* were cloned into pcDNA-dCas9-p300 Core vector (Addgene, Cambridge, MA; catalog number: 61357). sgRNA vector and *DNMT3A* or *TET2* vector were co-transfected into Hep3B cells. 48 hr later, gene expression was determined with q-PCR.

To prepare Hep3B cells stably expressing *Cas9* (*Koike-Yusa et al., 2014*), pspax2, pMD2G and pLentiCas9-BFP were co-transfected into HEK293T cells. Lentivirus particles were harvested 48 hr later and were used to infect Hep3B cells. Stable cell line was selected with Blasticidine (Sigma-Aldrich; catalog number: 15205; lot number: BCBM5270V). Targeting sequences corresponding to sgRNAs were synthesized (GENEWIZ), cloned into pKLV-U6gRNA(BbsI)-PGKpuro2ABFP vector (Addgene; catalog number: 50946), and packed into lentivirus particles to infect Hep3B cells stably expressing *Cas9*. Stable cell lines of *C/EBP-β* knockout were further selected with puromycin (Corning, NY; catalog number: 58-58-2; lot number: 61385051). The targeting sequences for *C/EBP-β* were: #1 5'-GGGCGCCTGGGGGCCGCCAA-3'; #2 5'-GGCGGCGGCGGCGGCGGGGG-3'.

## Luciferase reporter assay

The promoter fragment of *CRP* (−533 ~ +103 bp) was cloned into PGL4.10 (luc2) vector (Promega, Madison, WI; catalog number: E6651). Hep3B cells were transfected with 1.5 μg of PGL4.10 *CRP* reporter vector and 0.075 μg of phRL-TK (Promega; catalog number: E6241) using X-tremeGENE 9 DNA Transfection Reagent (Roche, Basel, Schweiz; catalog number: 06365787001; lot number: 23644700). After 48 hr of transfection, luciferase activities were measured using Dual-Luciferase Reporter Assay System (Promega; catalog number: E1960; lot number: 0000201344) on a Synergy HTX Multi-Mode Microplate Reader (BioTek, Winooski, VT). Activities of firefly luciferase were normalized with that of co-transfected Renilla luciferase.

## Chromatin immunoprecipitation (ChIP)

ChIP experiments were performed as described (*Nelson et al., 2006*). Briefly, Hep3B cells with or without transfected PGL4.10 vector containing *CRP* promoter were cross-linked with 1.42% formaldehyde at room temperature for 15 min. The reaction was stopped by addition of glycine to a final concentration of 125 mM. Cells were then sonicated in IP buffer (150 mM NaCl, 50 mM Tris, pH 7.5, 5 mM EDTA, 0.5 % NP-40, 1.0% Triton X-100) at 4°C for 10 min, followed by addition of anti-STAT3 (Santa Cruz, Dallas, TX; catalog number: sc-482X; lot number: B0615), anti-p50 (Santa Cruz; catalog number: sc-7178X; lot number: C0314) or anti-C/EBP-β (Santa Cruz; catalog number: sc-150X; lot number: J2215) for 15 min in ice/water bath with sonication. Protein-DNA complexes were isolated with nProtein A Sepharose 4 Fast Flow (GE Healthcare, Chicago, IL; catalog number: 17-5280-01; lot number: 10235150), and eluted with 10% Chelex 100 (Sigma-Aldrich; catalog number: C7901; lot number: SLBM2735V). The eluates were treated with proteinase K (BioFroxx, Hesse Einhausen, Germany; catalog number:1124MG100; lot number: 86081) and subjected to DNA purification, and the crosslinking was reversed by heating at 55°C for 30 min and boiling for 10 min. DNA was then purified and analyzed with q-PCR. The primer sequences for human *CRP* used were: genomic binding (Forward: 5'-CTCTTCCCGAAGCTCTGACACCT-3'; Reverse: 5'-AACAGCTTCTCCATGGTCACGTC-3'); vector binding (Forward: 5'-CTCTTCCCGAAGCTCTGACACCT-3'; Reverse: 5'-TGGCTTTACCAACAGTACCGGAT-3').

In some experiments, vectors were either entirely methylated with CpG methyltransferase (NEB, Ipswich, MA; catalog number: M0226L; lot number: 0311608) or site-specifically methylated at −53 or −108 CpG sites of *CRP* promoter through vector PCR using appropriately methylated primers before transfection.

## Statistical analysis

Data were presented as mean ± SEM. Statistical analysis was performed by the two-tailed Student's t-test, one-way ANOVA with Tukey post hoc or K-S tests as appropriate. Values of $p < 0.05$ were considered significant.

## Acknowledgements

We thank the Core Facility of School of Life Sciences, Lanzhou University for technical and instrumental support.

## Additional information

### Funding

| Funder | Grant reference number | Author |
|---|---|---|
| National Natural Science Foundation of China | 31671339 | Yi Wu |
| National Natural Science Foundation of China | 31870767 | Yi Wu |
| National Natural Science Foundation of China | 31570749 | Shang-Rong Ji |
| National Natural Science Foundation of China | 31770819 | Shang-Rong Ji |

The funders had no role in study design, data collection and interpretation, or the decision to submit the work for publication.

### Author contributions

Shi-Chao Zhang, Ming-Yu Wang, Jun-Rui Feng, Yue Chang, Investigation, Writing - review and editing; Shang-Rong Ji, Yi Wu, Supervision, Funding acquisition, Writing - original draft

## Author ORCIDs
Ming-Yu Wang (iD) https://orcid.org/0000-0002-6622-5996
Yi Wu (iD) https://orcid.org/0000-0003-0365-5590

## Ethics
Animal experimentation: The experiments conformed to the Guide for the Care and Use of Laboratory Animals published by NIH, and were conducted according to the protocols approved by the Ethics Committee of Animal Experiments of Xi'an Jiaotong University and Lanzhou University.

## Decision letter and Author response
Decision letter https://doi.org/10.7554/eLife.51317.sa1
Author response https://doi.org/10.7554/eLife.51317.sa2

# Additional files
## Supplementary files
- Supplementary file 1. Key resources table.

- Transparent reporting form

## Data availability
Sequencing data have been deposited in GEO under accession code GSE146797.

The following dataset was generated:

| Author(s) | Year | Dataset title | Dataset URL | Database and Identifier |
|---|---|---|---|---|
| Zhang S-C, Wang M-Y, Feng J-R, Chang Y, Ji S-R, Wu Y | 2020 | WGBS and RNA-seq of mouse livers and human Hep3B cells | https://www.ncbi.nlm.nih.gov/geo/query/acc.cgi?acc=GSE146797 | NCBI Gene Expression Omnibus, GSE146797 |

The following previously published datasets were used:

| Author(s) | Year | Dataset title | Dataset URL | Database and Identifier |
|---|---|---|---|---|
| Broad Institute | 2012 | Bisulfite-Seq analysis of WGBS_Lib 11 derived from human liver cells | https://www.ncbi.nlm.nih.gov/geo/query/acc.cgi?acc=GSM916049 | NCBI Gene Expression Omnibus, GSM916049 |
| Xin Li | 2016 | liver_N3_BS | https://www.ncbi.nlm.nih.gov/geo/query/acc.cgi?acc=GSM1716965 | NCBI Gene Expression Omnibus, GSM1716965 |
| UCSD AND SALK | 2013 | Whole Genome Shotgun Bisulfite Sequencing of Fat Cells from Human STL003 | https://www.ncbi.nlm.nih.gov/geo/query/acc.cgi?acc=GSM1120331 | NCBI Gene Expression Omnibus, GSM1120331 |
| UCSD AND SALK | 2013 | Whole Genome Shotgun Bisulfite Sequencing of Adrenal Cells from Human STL003 | https://www.ncbi.nlm.nih.gov/geo/query/acc.cgi?acc= GSM1120325 | NCBI Gene Expression Omnibus, GSM1120325 |
| UCSD AND SALK | 2013 | Whole Genome Shotgun Bisulfite Sequencing of Aorta Cells from Human STL003 | https://www.ncbi.nlm.nih.gov/geo/query/acc.cgi?acc=GSM1120329 | NCBI Gene Expression Omnibus, GSM1120329 |
| UCSD AND SALK | 2013 | Whole Genome Shotgun Bisulfite Sequencing of Esophagus Cells from Human STL003 | https://www.ncbi.nlm.nih.gov/geo/query/acc.cgi?acc=GSM983649 | NCBI Gene Expression Omnibus, GSM983649 |
| UCSD AND SALK | 2013 | Whole Genome Shotgun Bisulfite Sequencing of Gastric Cells from Human STL003 | https://www.ncbi.nlm.nih.gov/geo/query/acc.cgi?acc=GSM1120333 | NCBI Gene Expression Omnibus, GSM1120333 |
| UCSD AND SALK | 2013 | Whole Genome Shotgun Bisulfite Sequencing of Lung Cells from Human STL002 | https://www.ncbi.nlm.nih.gov/geo/query/acc.cgi?acc=GSM983647 | NCBI Gene Expression Omnibus, GSM983647 |

| UCSD AND SALK | 2013 | Whole Genome Shotgun Bisulfite Sequencing of Ovary Cells from Human STL002 | https://www.ncbi.nlm.nih.gov/geo/query/acc.cgi?acc=GSM1120323 | NCBI Gene Expression Omnibus, GSM1120323 |
|---|---|---|---|---|
| UCSD AND SALK | 2013 | Whole Genome Shotgun Bisulfite Sequencing of Psoas Cells from Human STL003 | https://www.ncbi.nlm.nih.gov/geo/query/acc.cgi?acc=GSM1010986 | NCBI Gene Expression Omnibus, GSM1010986 |
| UCSD AND SALK | 2013 | Whole Genome Shotgun Bisulfite Sequencing of Right Atrium Cells from Human STL003 | https://www.ncbi.nlm.nih.gov/geo/query/acc.cgi?acc=GSM1120335 | NCBI Gene Expression Omnibus, GSM1120335 |
| UCSD AND SALK | 2013 | Whole Genome Shotgun Bisulfite Sequencing of Sigmoid Colon Cells from Human STL001 | https://www.ncbi.nlm.nih.gov/geo/query/acc.cgi?acc=GSM983645 | NCBI Gene Expression Omnibus, GSM983645 |
| UCSD AND SALK | 2013 | Whole Genome Shotgun Bisulfite Sequencing of Spleen Cells from Human STL003 | https://www.ncbi.nlm.nih.gov/geo/query/acc.cgi?acc=GSM983652 | NCBI Gene Expression Omnibus, GSM983652 |
| UCSD AND SALK | 2013 | Whole Genome Shotgun Bisulfite Sequencing of Thymus Cells from Human STL001 | https://www.ncbi.nlm.nih.gov/geo/query/acc.cgi?acc=GSM1120322 | NCBI Gene Expression Omnibus, GSM1120322 |

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
