## [Decision Letter]

**Acceptance summary:**

The C reactive protein (*CRP*) belongs to the family of acute phase reactant genes, which play an essential role in inflammatory responses to infection or tissue injury in the liver. CPR needs to be quickly activated after stimulation by cytokines, and its activation is reversible upon resolution of inflammation. Zhang and colleagues report here that the dynamic expression of CPR is causally regulated by DNA methylation, and that this type of regulation may apply to other inducible genes that are endowed with CpG-poor promoters and have evolved CpG-containing binding sites for transcription factors specialized in rapid and reversible responses.

**Decision letter after peer review:**

Thank you for submitting your article "Reversible promoter methylation determines fluctuating expression of acute phase proteins" for consideration by *eLife*. Your article has been reviewed by two peer reviewers, and the evaluation has been overseen by a Reviewing Editor and Jessica Tyler as the Senior Editor. The reviewers have opted to remain anonymous.

The reviewers have discussed the reviews with one another and the Reviewing Editor has drafted this decision to help you prepare a revised submission.

Summary:

The interesting novelty and appeal of the study comes from 1) the biological importance of *CRP* and acute phase reactant genes in general, and 2) the details of *CRP* regulation in sequential phases by STAT3, NFkB and DNA demethylation/5OHT, and subsequent CEBP binding and activity. The evidence is supported by a satisfactory range of complementary experiments, from functional screens to modulation of DNA methylation by chemical inhibitors and knock-out or knock-down of methylating (*DNMT3A*) or demethylating (*TET2*) enzymes, with validations by targeted epigenomic editing.

However, the reviewers share the common concern that the manuscript is too concise in its current state to be fully instructive and convincing. More details are clearly required:

– In the Introduction, to provide proper background.

– In the Results section to explain the experimental design, methods and results and provide proper validation.

– In the Discussion, to replace the results in the current state of knowledge and clarify the model.

I agree that here is a need for further analyses of existing data and for inclusion of important missing controls to fully support interpretations and conclusions.

Please address the following comments in a revised version of your manuscript:

Essential revisions:

1) Provide information (in main text and figure legends) about i) how the CRISPR KO was performed: which exons were targeted? Was the analysis performed on single clones or cell bulk? And ii) how was DNA methylation analyzed in Figure 1: bisulfite sequencing? Available GEO datasets (and please cite the source if it is the case)?

2) Provide validation of *TET2* and *DNMT3A* KO or KD by Western blot and RT-qPCR, respectively (as a supplementary figure).

3) Provide evidence that DNA methylation of the *CRP* promoter is directly changed in the experiments of functional modulation of DNA methylation (using DNA methylation inhibitors, *DNMT3A* and *TET2* KO or KD). This is absolutely crucial for supporting conclusions. Along the same line, to further prove that the effect is not indirect, the expression of STAT3, p50 and C/EBP should be assessed (Western blot or RT-qPCR) in *DNMT3A* KO, *TET2* KO and upon treatment with DNA methylation inhibitors.

4) Throughout the manuscript, figures illustrating methylation profiles are needed. In Figure 1F, please show raw bisulfite sequencing data with individual clones rather than fold differences between the two alleles-an unusual and incomplete way to show the data. This will benefit interpretation of differences between the 286A and 286C alleles (are most of the maintained CpGme (Figure 1G) on the A allele?). Also, please show WGBS tracks of the *CRP* gene in various human tissues (available from public databases) to see how many CpG sites of the *CRP* promoter are specifically hypomethylated in liver.

5) At the end of the manuscript, the authors investigate if demethylation of CpG-poor promoters is a general feature for the activation of APR genes, in cells and liver. This is important and potentially interesting but again, the authors need to provide much more details on what was done and to perform a more thorough analysis of their WGBS and RNA-seq data. Otherwise, Figure 7 is a too vague genomic characterization of CpG frequency and expression associations. How many replicates of RNA-seq and WGBS? Do replicates correlate? How many genes upregulated and what are the genes upregulated? Is *CRP* demethylated upon inflammation in the liver (genome browser views)? How do changes in expression and methylation correlate? How many DMRs and do they preferentially locate to CpG poor promoters? These are obvious questions that need to be addressed, possibly in supplementary figures.

6) To further support their assumption that APR genes may behave in general as *CRP*, it would be beneficial if the authors could look by RT-qPCR at expression of several of these genes in the +/- 5aza conditions (with cDNA from Figure 2A) to see if they go up, as they would predict. They could also include some gene sets based on PRG, CpG low status, etc… as well as known rapid response genes (see Hargreaves et al., 2009) that seem to be regulated in a very different manner than APR genes. These may be similar to the authors "cytokine" genes.

7) The sequential model proposed by the authors is confusing. First, according to Figure 4 and Figure 5B-D, all three TFs (STAT3, p50 and C/EBP) are potentially sensitive to DNA methylation, which contradicts the model. Second, to test which TF requires demethylation for binding, the kinetic experiments described in Figure 5B-D should be performed in *TET2* KO cells (also generated by the authors) instead of *DNMT3A* KO cells, to determine which TF has reduced binding when demethylation is compromised.

8) There is a lack of thorough introduction and/or discussion of existing evidence from the literature to place the authors' study in the current knowledge context. First, it is known that dynamic DNA methylation changes usually affect CpG-poor promoters, in part due to lack of protective binding of methyl DNA binding proteins, while CpG islands (CGI) promoters, on the contrary, are predominantly non-methylated and stable to ensure expression of housekeeping genes and primary response genes (Hargreaves et al., 2009). Apart from the studies cited in the Discussion of this manuscript, acute changes in DNA methylation have been described in other contexts such as in neurons, upon activation (Guo et al., 2011 among others). As in the present study, but at a genome wide scale, the authors described dynamic reversible methylation of low density CpG promoters in brain-specific genes, and correlated them with gene expression. Moreover, CpG methylation is known to affect transcription factor binding (Yin et al., 2017), and specifically STAT3 binding (Thomas, 2012). And very recently, in neurons as well, TF were described to recruit *TET1* to demethylate specific promoters upon neuronal activation (Sun et al., 2019). Second, in relationship with inflammation and APR gene regulation, the authors should discuss the work of Stephen Smale (Hargreaves et al., 2009) -and potentially others- about conventional categorization of CpG-rich promoters in inflammatory gene induction: nucleosome poor promoters, accessible, independent of remodeling, one step induction. This provides an interesting contrast with the sequential kinetics of *CRP*, regulated in part by changes in DNA methylation and temporal waves of TF binding, as demonstrated here by the authors.

Recommended changes:

1) Why is *CRP* hypomethylated in liver cells in unstimulated conditions? Is it because a basal level of expression is needed? This could be discussed.

2) In Figure 2A, the authors show that DNA methylation inhibitors enhance *CRP* expression at the resting stage. However, they did not assess as to whether *CRP* expression is modified by these inhibitors after induction (it should not), while this would be an important point for their demonstration. Please include this analysis after Figure 2A.

3) To assess whether DNA demethylation precedes or follows gene activation, please add a 24h time point for DNA methylation analysis (Figure 1G), if possible.

---

## [Author Response]

Essential revisions:1) Provide information (in main text and figure legends) about i) how the CRISPR KO was performed: which exons were targeted? Was the analysis performed on single clones or cell bulk? And ii) how was DNA methylation analyzed in Figure 1: bisulfite sequencing? Available GEO datasets (and please cite the source if it is the case)?

In CRISPR KO experiments, exon 3 or 7 of *TET2* and exon 14 or 2 of *DNMT3A* were targeted. The analysis was performed on cell bulk in Figure 3 and 5, and was performed on single clones in Figure 6.

In Figure 1C, DNA methylation levels of *CRP* promoter in different human tissues were obtained from available GEO datasets generated by whole-genome bisulfite sequencing: Liver 01-GSM916049, Liver 02-GSM1716965, Adipose-GSM1120331, Adrenal-GSM1120325, Aorta-GSM1120329, Esophagus-GSM983649, Gaster-GSM1120333, Lung-GSM983647, Ovary-GSM1120323, Muscle-GSM1010986, Atrium-GSM1120335, Colon-GSM983645, Spleen-GSM983652, Thymus-GSM1120322. In other figures of Figure 1, DNA methylation levels of *CRP* promoter were determined with bisulfite cloning sequencing.

These descriptions have been included in the revised manuscript.

2) Provide validation of TET2 and DNMT3A KO or KD by Western blot and RT-qPCR, respectively (as a supplementary figure).

Immunoblotting and qPCR validation of *TET2/DNMT3A* KD/KO showed that the levels of *TET2* or *DNMT3A* were effectively reduced (Figure 3—figure supplement 2).

3) Provide evidence that DNA methylation of the CRP promoter is directly changed in the experiments of functional modulation of DNA methylation (using DNA methylation inhibitors, DNMT3A and TET2 KO or KD). This is absolutely crucial for supporting conclusions. Along the same line, to further prove that the effect is not indirect, the expression of STAT3, p50 and C/EBP should be assessed (Western blot or RT-qPCR) in DNMT3A KO, TET2 KO and upon treatment with DNA methylation inhibitors.

We have performed additional experiments as suggested by the editor and reviewers. We have assessed the changes in DNA methylation of *CRP* promoter in Hep3B cells following treatment with DNA methylation inhibitor 5-aza, or *DNMT3A/TET2* KD/KO. The results indicated that 5-aza and *DNMT3A* KD/KO reduced, whereas *TET2* KD/KO augmented DNA methylation of *CRP* promoter (Figure 2—figure supplement 1 and Figure 3—figure supplement 1).

We have assessed the expression of STAT3, p50 and C/EBP-β in Hep3B cells following treatment with 5-aza, or *DNMT3A/TET2* KD/KO (Figure 3—figure supplement 3). 5-aza treatment reduced the expression of STAT3, but did not affect that of p50 and C/EBP-β. *DNMT3A* or *TET2* KD showed little effect on the expression of the three TFs. *DNMT3A* or *TET2* KO tended to reduce their expression. Because the observed effects on *CRP* expression by 5-aza or *DNMT3A/TET2* KD/KO are consistent or complementary, the above results suggest that these manipulations regulate *CRP* expression by directly modulating promoter methylation.

Further support to this suggestion is provided by specifically modulating promoter methylation of *CRP* via dCas9-mediated targeting of the catalytic domains of DNMT3A or *TET2*. Enforced methylation of *CRP* promoter by *DNMT3A-dCas9* reduced the expression of *CRP* in Hep3B cells, but showed little effect on that of *serum amyloid A (SAA*, another major human APR) and *serum amyloid P component* (*SAP*, a paralog of *CRP*) (Figure 3F). By contrast, enforced demethylation of *CRP* promoter by *TET2*-dCas9 only selectively enhanced the expression of *CRP* (Figure 3G). We thus conclude that *DNMT3A* and *TET2*-tuned methylation status of *CRP* promoter constitutes a key part of the regulatory mechanism that causally determines the expression.

4) Throughout the manuscript, figures illustrating methylation profiles are needed. In Figure 1F, please show raw bisulfite sequencing data with individual clones rather than fold differences between the two alleles-an unusual and incomplete way to show the data. This will benefit interpretation of differences between the 286A and 286C alleles (are most of the maintained CpGme (Figure 1G) on the A allele?). Also, please show WGBS tracks of the CRP gene in various human tissues (available from public databases) to see how many CpG sites of the CRP promoter are specifically hypomethylated in liver.

Figure 1 has been carefully revised according to the comments of the editor and reviewers. Raw bisulfite sequencing data with individual clones are shown in Figure 1F. The results indicate that most maintained CpGme were on -286C allele. The WGBS tracks of *CRP* gene are shown in Figure 1C. -53, -108, -286 and -306CpG of *CRP* are all most demethylated in human livers with only -255CpG in Liver 01 being more methylated than in Adrenal.

5) At the end of the manuscript, the authors investigate if demethylation of CpG-poor promoters is a general feature for the activation of APR genes, in cells and liver. This is important and potentially interesting but again, the authors need to provide much more details on what was done and to perform a more thorough analysis of their WGBS and RNA-seq data. Otherwise, Figure 7 is a too vague genomic characterization of CpG frequency and expression associations. How many replicates of RNA-seq and WGBS? Do replicates correlate? How many genes upregulated and what are the genes upregulated? Is CRP demethylated upon inflammation in the liver (genome browser views)? How do changes in expression and methylation correlate? How many DMRs and do they preferentially locate to CpG poor promoters? These are obvious questions that need to be addressed, possibly in supplementary figures.

We have performed additional experiments and analysis as suggested by the editor and reviewers.

Two biological replicates were examined for both RNA-seq and WGBS. For RNA-seq, the correlations (R^2^) between the replicates are 0.991 for resting Hep3B cells, 0.998 for induced Hep3B cells, 0.884 for resting mouse livers and 0.803 for induced mouse livers (Figure 7—figure supplement 1A and B). For WGBS, the correlations (R^2^) between the replicates are 0.828 for resting mouse livers and 0.810 for induced mouse livers (Figure 7—figure supplement 1C).

The expression of 2705 and 3371 genes were changed by over 2-fold (up- and down-regulated) in induced Hep3B cells and in induced mouse livers, respectively. In Hep3B cells, the “regulation of inflammatory response”, “acute inflammatory response” and “regulation of immune effector process” are the top 3 activated biological processes. In induced mouse livers, the “acute inflammatory response”, “positive regulation of response to external stimulus” and “cell chemotaxis” are the top 3 activated biological processes (Figure 7—figure supplement 1D and E).

In contrast to human *CRP*, mouse *CRP* is not an acute phase protein [^1-3^] (possibly due to the loss of binding sites for C/EBP-β as revealed by bioinformatics analysis), and its promoter did not undergo demethylation in induced livers (Author response image 1). Although genes with CpG-poor promoters tended to show greater changes in both expression and methylation (Figure 7D and 7E), changes in expression and promoter methylation per se did not show significant correlation. These would suggest that promoter methylation and gene expression may be qualitatively but not quantitatively associated with each other.

Overall, 1262 differentially methylated regions (DMR; 1k sequence window) were identified in induced versus resting mouse livers. The CpG ratios of these DMRs (median: 0.07; 25th~75th percentile: 0.04~0.17) are significantly lower than that of non-DMRs (median: 0.16; 25th~75th percentile: 0.11~0.23; *p* < 10^-15^). Therefore, differential methylation prefers to occur in CpG-poor regions. However, there are only 85 promoters located in these DMRs. Such an under-representation is consistent with previous findings made in mouse neurons activated by electroconvulsive stimulation ^4^.

**Author response image 1. respfig1:** The bisulfite sequencing tracks of *Nxpe5* (Log2(FC of expression) = 3.6), *Cyp4a14* (Log2(FC of expression) = -3.1), *Clec4n* (Log2(FC of expression) = 1.9) and *CRP* (Log2(FC of expression) = 0.8). Promoter CpGs are boxed by red rectangles.

6) To further support their assumption that APR genes may behave in general as CRP, it would be beneficial if the authors could look by RT-qPCR at expression of several of these genes in the +/- 5aza conditions (with cDNA from Figure 2A) to see if they go up, as they would predict. They could also include some gene sets based on PRG, CpG low status, etc… as well as known rapid response genes (see Hargreaves et al., 2009) that seem to be regulated in a very different manner than APR genes. These may be similar to the authors "cytokine" genes.

We have performed additional experiments as suggested by the editor and reviewers. The effects of 5-aza treatment on the expression of 2 additional APRs with CpG-poor promoters, 3 class A PRGs with CpG-rich promoters, 4 class B PRGs with CpG-poor promoters, and 2 class B SRG with CpG-poor promoters in Hep3B cells were determined with q-PCR (Author response image 2). 5-aza treatment significantly enhanced most of the examined genes with CpG-poor promoters but not those with CpG-rich promoters. Therefore, PRGs with CpG-rich promoters are likely induced by mechanisms independent of promoter methylation.

**Author response image 2. respfig2:** Expression of 2 additional APRs with CpG-poor promoters (*SAA* and *LIF*), 3 class A PRGs with CpG-rich promoters (*CXCL2, Nfkbia, Pim1*), 4 class B PRGs with CpG-poor promoters (*TNF, CSF2, IL1b, IL23a*), and 2 class B SRG with CpG-poor promoters (*LCN2* and *MX2*) in Hep3B cells treated with or without 5-aza for 36 h. Of note, *TNF* has been classified as a Class A PRG in mouse macrophages ^5^. However, the observed/expected CpG ratio of *TNF* promoter in human genome is 0.31, we thus classified it as a class B PRG here.

7) The sequential model proposed by the authors is confusing. First, according to Figure 4 and Figure 5B-D, all three TFs (STAT3, p50 and C/EBP) are potentially sensitive to DNA methylation, which contradicts the model. Second, to test which TF requires demethylation for binding, the kinetic experiments described in Figure 5B-D should be performed in TET2 KO cells (also generated by the authors) instead of DNMT3A KO cells, to determine which TF has reduced binding when demethylation is compromised.

We have performed additional experiments as suggested by the editor and reviewers. *TET2* KO almost abrogated the recruitment of p50 and C/EBP-β to *CRP* promoter, but the recruitment of STAT3 was still apparent (Figure 5A-D). This indicates that STAT3 can nevertheless be recruited to *CRP* promoter even when demethylation is compromised, consistent with the observations that vector binding of STAT3 was least sensitive to methylation (Figure 4B and C). This also indicates that STAT3 can act largely independent of p50 and C/EBP-β to drive *CRP* expression, albeit markedly impaired (Figure 5A). Indeed, the sole activation of STAT3 in wildtype cells drove *CRP* expression to a level comparable to the minor peak, whereas the sole activation of NF-κB was completely ineffective (Figure 5E). NF-κB inhibition (with intact STAT3) at the induced state, however, resulted in an even stronger methylation of *CRP* promoter (Figure 5F) and a reversal of allelic imbalance (Figure 5G). Therefore, promoter demethylation requires p50 that acts downstream of STAT3.

Interestingly, enforced DNA demethylation by *DNMT3A* KO not only tripled *CRP* expression during the entire course of induction, but eliminated the time lag between the two waves (Figure 5A). The augmented amplitude can be explained by enhanced recruitment of three TFs, while the altered dynamics may correspond to the shifted timing of p50 recruitment (Figure 5B-D). As such, p50 selectively recruited during the time lag could be responsible for promoter demethylation to prime the second wave. Accordingly, C/EBP-β appears to be the major effector that responds to promoter demethylation: its overexpression did not demethylate *CRP* promoter (Figure 5H and I), but when combined with blockage of DNA methylation, it drove the resting expression of *CRP* to a level approaching to that induced by IL-6 and IL-1β (Figure 5J). *C/EBP-β* KO, however, lowered the induced expression of *CRP* by ~70% (Figure 5K). These results together are consistent with the proposed sequential model.

We have carefully revised the manuscript to elaborate the sequential model.

8) There is a lack of thorough introduction and/or discussion of existing evidence from the literature to place the authors' study in the current knowledge context. First, it is known that dynamic DNA methylation changes usually affect CpG-poor promoters, in part due to lack of protective binding of methyl DNA binding proteins, while CpG islands (CGI) promoters, on the contrary, are predominantly non-methylated and stable to ensure expression of housekeeping genes and primary response genes (Hargreaves et al., 2009). Apart from the studies cited in the Discussion of this manuscript, acute changes in DNA methylation have been described in other contexts such as in neurons, upon activation (Guo et al., 2011 among others). As in the present study, but at a genome wide scale, the authors described dynamic reversible methylation of low density CpG promoters in brain-specific genes, and correlated them with gene expression. Moreover, CpG methylation is known to affect transcription factor binding (Yin et al., 2017), and specifically STAT3 binding (Thomas, 2012). And very recently, in neurons as well, TF were described to recruit TET1 to demethylate specific promoters upon neuronal activation (Sun et al., 2019). Second, in relationship with inflammation and APR gene regulation, the authors should discuss the work of Stephen Smale (Hargreaves et al., 2009) -and potentially others- about conventional categorization of CpG-rich promoters in inflammatory gene induction: nucleosome poor promoters, accessible, independent of remodeling, one step induction. This provides an interesting contrast with the sequential kinetics of CRP, regulated in part by changes in DNA methylation and temporal waves of TF binding, as demonstrated here by the authors.

The manuscript has been carefully revised according to the comments of the editor and reviewers.

Acute changes in DNA methylation have also been analyzed at a genome-wide scale in previous studies. One study examined mouse neurons activated by electroconvulsive stimulation, and found that methylation changes were preferentially occurred in CpG-poor regions [^4^]. Though promoters were underrepresented in these regions, their methylation changes were nevertheless modestly anticorrelated with gene expression [^4^]. Such CpG content-dependent changes in DNA methylation, however, were not observed in another study examining neurons activated by contextual learning [^6^]. Moreover, DNA methylation changes in human dendritic cells following infection rarely occurred at promoters [^7, 8^], and were claimed to be a consequence of gene expression [^7^]. Those findings thus argue that the regulation of reversible methylation on inducible expression of genes with CpG-poor promoters may be context-dependent, and that rigorously controlled case study should be integrated into genome-wide investigation to conclude on causality.

Interestingly, genes showing inducible expression in macrophages activated by endotoxin have been classified into two groups: nucleosome remodeling-independent genes with CpG-rich promoters, and nucleosome remodeling-dependent genes with CpG-poor promoters [^5^]. Despite their distinct requirements for SWI/SNF complexes [^5^], preassembled Pol II [^9^] and new protein synthesis [^5, 9, 10^], both groups of genes exhibit a single-wave kinetics of induction [^5, 9, 10^]. This is in contrast with *CRP* and probably other major APRs, which show a two-wave kinetics of induction with the second wave licensed by promoter demethylation. However, nucleosome remodeling may also act downstream in the second wave, as C/EBP-β has recently been shown to promote *CRP* expression by recruiting BRG1 via MKL1 [^11^]. Conversely, promoter methylation may also contribute to inducible expression of nucleosome remodeling-dependent genes in macrophages by directly regulating TF recruitment [^12-14^].

Despite that most CpGs in mammalian promoters not associated with CGIs are usually methylated [^15, 16^], *TETs* and *DNMTs* can nevertheless mediate active demethylation and de novomethylation, respectively. This indicates that the methylation status of part of the genome may depend on the balance of the two types of enzymes [^15-19^]. Cellular signaling able to tip the balance of *TETs* and *DNMTs* could thus represent a regulatory mechanism that finely and reversibly tunes the expression of certain genes in response to environmental cues. In case of *CRP, DNMT3A* and *TET2* appear to be involved in the regulation. Though the mechanism of TF-induced local demethylation is not fully understood [^16^], TFs including NF-κB and EGR1 have been reported to evoke DNA demethylation in neurons [^20^] for example by recruiting *TET1* [^21^]. Future study is warranted to elucidate how TFs regulate the balance of *TETs* and *DNMTs* and to discover scenarios where regulation by reversible DNA methylation plays a prominent role.

These discussions have been included in the revised manuscript.

Recommended changes:1) Why is CRP hypomethylated in liver cells in unstimulated conditions? Is it because a basal level of expression is needed? This could be discussed.

As pointed by the editor and reviewers, hepatocytes secrete *CRP* in unstimulated conditions, yielding a basal circulating level of about 1~3 μg/ml in apparently healthy individuals [^22-24^]. *CRP* has been proposed to act as a soluble pattern recognition receptor in host defense [^25, 26^]. Therefore, basal levels of circulating *CRP* may constitute an integral part of immune surveillance that contributes to the silent clearance of invading pathogens and apoptotic cells in homeostasis. These discussions have been included in the revised manuscript.

2) In Figure 2A, the authors show that DNA methylation inhibitors enhance CRP expression at the resting stage. However, they did not assess as to whether CRP expression is modified by these inhibitors after induction (it should not), while this would be an important point for their demonstration. Please include this analysis after Figure 2A.

We have performed additional experiments as suggested by the editor and reviewers. Treating Hep3B cells with 5-aza significantly enhanced *CRP* expression at the resting but not at the induced state (Figure 2A) wherein *CRP* promoter underwent active demethylation (Figure 1G).

3) To assess whether DNA demethylation precedes or follows gene activation, please add a 24h time point for DNA methylation analysis (Figure 1G), if possible.

This experiment was originally arranged to be performed following those assessing methylation changes induced by 5-aza, *TET/DNMT* KO or KD (Figure II). Before its completion, however, our lab had to be closed due to the SARS-CoV-2 epidemic. Therefore, we could not arrange this experiment in a reasonable time frame. Nevertheless, the results shown in Figure 5 suggest that the demethylation of *CRP* promoter might follow the minor peak driven by STAT3, but precede the major peak driven by C/EBP-β.

References:

1) Baltz ML, de Beer FC, Feinstein A, Munn EA, Milstein CP, Fletcher TC, March JF, Taylor J, Bruton C, Clamp JR, Davies AJ, Pepys MB. Phylogenetic aspects of c-reactive protein and related proteins. Ann N Y Acad Sci. 1982;389:49-75

2) Pepys MB, Hirschfield GM. C-reactive protein: A critical update. J Clin Invest. 2003;111:1805-1812

3) Pathak A, Agrawal A. Evolution of c-reactive protein. Front Immunol. 2019;10:943

4) Guo JU, Ma DK, Mo H, Ball MP, Jang MH, Bonaguidi MA, Balazer JA, Eaves HL, Xie B, Ford E, Zhang K, Ming GL, Gao Y, Song H. Neuronal activity modifies the DNA methylation landscape in the adult brain. Nat Neurosci. 2011;14:1345-1351

5) Ramirez-Carrozzi VR, Braas D, Bhatt DM, Cheng CS, Hong C, Doty KR, Black JC, Hoffmann A, Carey M, Smale ST. A unifying model for the selective regulation of inducible transcription by cpg islands and nucleosome remodeling. Cell. 2009;138:114-128

6) Halder R, Hennion M, Vidal RO, Shomroni O, Rahman RU, Rajput A, Centeno TP, van Bebber F, Capece V, Garcia Vizcaino JC, Schuetz AL, Burkhardt S, Benito E, Navarro Sala M, Javan SB, Haass C, Schmid B, Fischer A, Bonn S. DNA methylation changes in plasticity genes accompany the formation and maintenance of memory. Nat Neurosci. 2016;19:102-110

7) Pacis A, Mailhot-Leonard F, Tailleux L, Randolph HE, Yotova V, Dumaine A, Grenier JC, Barreiro LB. Gene activation precedes DNA demethylation in response to infection in human dendritic cells. Proc Natl Acad Sci U S A. 2019;116:6938-6943

8) Pacis A, Tailleux L, Morin AM, Lambourne J, MacIsaac JL, Yotova V, Dumaine A, Danckaert A, Luca F, Grenier JC, Hansen KD, Gicquel B, Yu M, Pai A, He C, Tung J, Pastinen T, Kobor MS, Pique-Regi R, Gilad Y, Barreiro LB. Bacterial infection remodels the DNA methylation landscape of human dendritic cells. Genome Res. 2015;25:1801-1811

9) Hargreaves DC, Horng T, Medzhitov R. Control of inducible gene expression by signal-dependent transcriptional elongation. Cell. 2009;138:129-145

10) Ramirez-Carrozzi VR, Nazarian AA, Li CC, Gore SL, Sridharan R, Imbalzano AN, Smale ST. Selective and antagonistic functions of swi/snf and mi-2beta nucleosome remodeling complexes during an inflammatory response. Genes Dev. 2006;20:282-296

11) Fan Z, Li N, Xu Z, Wu J, Fan X, Xu Y. An interaction between mkl1, brg1, and c/ebpbeta mediates palmitate induced crp transcription in hepatocytes. Biochim Biophys Acta Gene Regul Mech. 2019;1862:194412

12) Thomas RM, Sai H, Wells AD. Conserved intergenic elements and DNA methylation cooperate to regulate transcription at the il17 locus. *J Biol Chem*. 2012;287:25049-25059

13) Hu S, Wan J, Su Y, Song Q, Zeng Y, Nguyen HN, Shin J, Cox E, Rho HS, Woodard C, Xia S, Liu S, Lyu H, Ming GL, Wade H, Song H, Qian J, Zhu H. DNA methylation presents distinct binding sites for human transcription factors. *ELife*. 2013;2:e00726

14) Yin Y, Morgunova E, Jolma A, Kaasinen E, Sahu B, Khund-Sayeed S, Das PK, Kivioja T, Dave K, Zhong F, Nitta KR, Taipale M, Popov A, Ginno PA, Domcke S, Yan J, Schubeler D, Vinson C, Taipale J. Impact of cytosine methylation on DNA binding specificities of human transcription factors. *Science*. 2017;356

15) Wu H, Zhang Y. Reversing DNA methylation: Mechanisms, genomics, and biological functions. *Cell*. 2014;156:45-68

16) Luo C, Hajkova P, Ecker JR. Dynamic DNA methylation: In the right place at the right time. *Science*. 2018;361:1336-1340

17) Jones PA. Functions of DNA methylation: Islands, start sites, gene bodies and beyond. *Nat Rev Genet*. 2012;13:484-492

18) Dor Y, Cedar H. Principles of DNA methylation and their implications for biology and medicine. *Lancet*. 2018;392:777-786

19) Blattler A, Farnham PJ. Cross-talk between site-specific transcription factors and DNA methylation states. *J Biol Chem*. 2013;288:34287-34294

20) Jarome TJ, Butler AA, Nichols JN, Pacheco NL, Lubin FD. Nf-kappab mediates gadd45beta expression and DNA demethylation in the hippocampus during fear memory formation. *Front Mol Neurosci*. 2015;8:54

21) Sun Z, Xu X, He J, Murray A, Sun MA, Wei X, Wang X, McCoig E, Xie E, Jiang X, Li L, Zhu J, Chen J, Morozov A, Pickrell AM, Theus MH, Xie H. Egr1 recruits tet1 to shape the brain methylome during development and upon neuronal activity. *Nat Commun*. 2019;10:3892

22) Ridker PM, Rifai N, Rose L, Buring JE, Cook NR. Comparison of c-reactive protein and low-density lipoprotein cholesterol levels in the prediction of first cardiovascular events. *N Engl J Med*. 2002;347:1557-1565

23) Danesh J, Wheeler JG, Hirschfield GM, Eda S, Eiriksdottir G, Rumley A, Lowe GD, Pepys MB, Gudnason V. C-reactive protein and other circulating markers of inflammation in the prediction of coronary heart disease. *N Engl J Med*. 2004;350:1387-1397

24) Zacho J, Tybjaerg-Hansen A, Jensen JS, Grande P, Sillesen H, Nordestgaard BG. Genetically elevated c-reactive protein and ischemic vascular disease. *N Engl J Med*. 2008;359:1897-1908

25) Bottazzi B, Doni A, Garlanda C, Mantovani A. An integrated view of humoral innate immunity: Pentraxins as a paradigm. *Annu Rev Immunol*. 2010;28:157-183

26) Du Clos TW. Pentraxins: Structure, function, and role in inflammation. *ISRN Inflamm*. 2013;2013:379040